# Primordial-material preservation and Earth lower-mantle structure: the influence of recycled oceanic crust

Matteo Desiderio[1], Anna J. P. Gülcher[2], and Maxim D. Ballmer[1]

[1]University College London, London, United Kingdom
[2]University of Bern, Bern, Switzerland

**Correspondence:** Matteo Desiderio (matteo.desiderio.20@ucl.ac.uk)

**Abstract.** The compositional structure of the Earth's lower mantle holds the key to understand the evolution of the coupled interior-atmosphere system, but remains elusive. Geochemical observations point to long-term preservation of primordial materials somewhere in the lower mantle, but the relationship of these reservoirs to geophysical anomalies is still debated. It has been shown that bridgmanitic material formed during magma-ocean crystallization can resist convective entrainment over geologic timescales to be preserved as "Bridgmanite-Enriched Ancient Mantle Structures" (BEAMS). BEAMS may host primordial geochemical reservoirs, but their style of preservation needs further testing. Using global-scale geodynamic models, we here explore how the physical properties of recycled oceanic crust (ROC) affect the style of primordial-material preservation. We show that significant BEAMS preservation is only obtained for ROC accumulation in the deep mantle as thermochemical piles, or a global ROC layer, due to high intrinsic ROC density. High intrinsic ROC viscosity also enhances BEAMS preservation, especially in the thermochemical piles regime. We find that primordial and recycled domains have a mutually protective effect. The coupled preservation of BEAMS-like structures in the mid-mantle and ROC piles in the lowermost mantle is consistent with the diverse isotopic record of ocean-island basalts, reconciling the preservation of distinct geochemical reservoirs in a vigorously convecting mantle.

## 1   Introduction

Heterogeneities pervade Earth's Lower Mantle (LM). The origin of LM thermochemical structure and its relation to convection patterns is key to understand heat and material fluxes through the mantle, and how these may impact the evolution of the planet as a whole. The geochemical record hints at the long-term preservation of chemically distinct reservoirs. Ocean Island Basalts (OIBs), with their diverse range of isotopic compositions, carry the signatures of both recent and ancient recycled materials (Hofmann, 1997; Allègre and Turcotte, 1986; White, 2015), as well as of primitive differentiation events that occurred in the immediate aftermath of Earth's accretion (Caracausi et al., 2016; Mukhopadhyay, 2012; Porcelli and Elliott, 2008; Touboul et al., 2012; Rizo et al., 2016; Mundl et al., 2017; Jackson et al., 2017). Classic travel time seismic tomography models, in turn, offer a present-day snapshot of the LM, many demonstrating whole-mantle convection and suggesting efficient mixing between layers in the mantle (Fukao and Obayashi, 2013; Houser et al., 2008; Ritsema et al., 2011). Recent global full-waveform inversion tomography, on the other hand, shows a more complex, heterogeneous mantle structure than traditionally

imaged (Schouten et al., 2024; Thrastarson et al., 2024). The most striking recurring features in all global seismic tomography models are the two Large Low Shear Velocity Provinces (LLSVPs) (Dziewonski, 1984; Cottaar and Lekic, 2016), continent-sized slow anomalies of unclear nature (Garnero et al., 2016; McNamara, 2019): recent observations indicate that LLSVPs are indeed compositionally distinct domains (e.g., Moulik and Ekström, 2016; Richards et al., 2023; Lau et al., 2017), although whether they represent recycled and/or primordial geochemical reservoirs remains unclear (e.g., Tackley, 2012). Integrating geochemical and geophysical perspectives is thus crucial to better understand the structure and dynamics of the lower mantle.

As typical for terrestrial planets, Earth endured at least one phase of global-scale melting with a magma ocean that likely reached down to the Core Mantle Boundary (CMB) due to massive post-accretion and differentiation energy releases (e.g., Solomatov, 2015; Nakajima and Stevenson, 2015; Fischer et al., 2017). The crystallization of this magma ocean likely culminated with a molten layer confined above the CMB (the basal magma ocean, or BMO) (Labrosse et al., 2007; Boukaré et al., 2025), which has been put forward as an ideal reservoir for noble gases (Coltice et al., 2011; Li et al., 2022; Ozgurel and Caracas, 2023; Jackson et al., 2021) and other primordial isotopic signatures (Boyet and Carlson, 2005; Rizo et al., 2016; Caracas et al., 2019). Assuming a pyrolite-like starting composition of the magma ocean, melting experiments confirm the mineral Bridgmanite (Bm) as the liquidus phase at LM pressures, leading to the crystallization of massive bridgmanitic domains (Boukaré et al., 2015; Nabiei et al., 2021). Geodynamic modeling shows that these Bridgmanite-Enriched Ancient Mantle Structures (BEAMS) may be preserved in the modern mantle (Ballmer et al., 2017a; Gülcher et al., 2020; Gülcher et al., 2021; Manga, 1996; Becker et al., 1999), owing to the high intrinsic viscosity of Bm (Girard et al., 2016; Tsujino et al., 2022; Okamoto and Hiraga, 2024). Progressive iron-enrichment in later-stage BMO cumulates (Nabiei et al., 2021; Caracas et al., 2019) would instead lead to accumulation over the Core Mantle Boundary (CMB) (Labrosse et al., 2007; Boukaré et al., 2025). In this scenario, LLSVPs represent thermochemical piles of dense primordial material, supported by geodynamic models (Gülcher et al., 2021; Li et al., 2014b; Le Bars and Davaille, 2004; Li and McNamara, 2018), and consistent with seismic and mineral physics constraints (Ballmer et al., 2017b; Vilella et al., 2021; Deschamps et al., 2012). On the other hand, it has been proposed that LLSVPs incorporate Recycled Oceanic Crust (ROC) (Tackley, 2012; Li et al., 2014b; White, 2015; Christensen and Hofmann, 1994; Brandenburg and Van Keken, 2007), since its high intrinsic density facilitates segregation and accumulation over the CMB (Ricolleau et al., 2010; Hirose et al., 2005). Tomographic images of deep-subducted lithosphere (Van Der Hilst et al., 1997; Fukao and Obayashi, 2013; Ritsema et al., 2011; Grand, 1994; van der Meer et al., 2010) confirm that oceanic crust is recycled to the lower mantle. Experimental evidence (Tschauner et al., 2021; Thomson et al., 2019; Gréaux et al., 2019) and numerical models (Li et al., 2014b; Jones et al., 2020; Yan et al., 2020; Christensen and Hofmann, 1994; Brandenburg and Van Keken, 2007; Panton et al., 2023) support the view that LLSVPs are made up of this recycled material, and possibly tapped by mantle plumes (French and Romanowicz, 2015; Burke et al., 2008; Doubrovine et al., 2016; Heyn et al., 2020).

However, several aspects of present-day mantle heterogeneity remain poorly understood. In particular, ROC physical properties depend on its composition, which may vary over time (Nakagawa et al., 2010; Herzberg et al., 2010). ROC intrinsic density critically determines segregation efficiency from the subducted lithosphere, the stability of piles and their shape (Yan et al., 2020; Tackley, 2011; Tan and Gurnis, 2007; Christensen and Hofmann, 1994). Moreover, while the viscosity of deep-

subducted ROC and other potential pile-forming materials remains largely unconstrained (Immoor et al., 2022; Miyagi et al., 2009; Marquardt and Thomson, 2020; Okamoto and Hiraga, 2024), several geodynamic studies indicate that this property determines the stability, shape, and thermochemical structure of the LLSVPs (e.g., Bower et al., 2013; Citron et al., 2020; Heyn et al., 2018; Mulyukova et al., 2015; Li and McNamara, 2018; Desiderio and Ballmer, 2024; Dannberg et al., 2023). Previous work by Gülcher et al. (2021) investigated the coexistence of recycled and primordial heterogeneities in the mantle, suggesting a stabilizing feedback between LLSVPs and bridgmanite-rich primordial domains. Yet, while Desiderio and Ballmer (2024) show that the intrinsic viscosity of pile-forming materials controls CMB heat flux, mantle thermal evolution, and convective vigour, no previous study has systematically explored the effects of ROC properties on the preservation of BEAMS.

Here, we apply global-scale 2D numerical models of mantle convection to systematically investigate the influence of ROC intrinsic LM density and LM viscosity on the long-term thermochemical evolution of Earth's mantle. Our models include an intrinsically strong primordial material, initially located in the lowermost mantle, and self-consistent formation of basaltic crust via partial melting. This setup allows us to explore the dynamic interaction between self-consistently formed (recycled) thermochemical piles and mid-mantle (primitive) viscous domains. We find that large BEAMS-like heterogeneities are only preserved when ROC density is sufficient to stabilize basaltic piles (or a continuous layer) above the CMB. Moreover, ROC viscosity controls thermochemical gradients in the lowermost mantle, further enhancing BEAMS preservation. We put our model predictions in context with geophysical constraints available for modern Earth. Finally, we examine implications in terms of early Earth history and mantle geochemistry.

## 2  Methods

We apply the finite-volume code StagYY (Tackley, 2008) to model thermochemical convection of the mantle in 2D spherical-annulus geometry. We use the anelastic compressible liquid approximation with infinite Prandtl number to solve for the conservation of mass, energy and momentum. As in Gülcher et al. (2021); Desiderio and Ballmer (2024), the model domain consists of a grid of $512 \times 96$ cells, horizontally and vertically respectively. Vertical grid refinement occurs around the depth of 660 km and towards the lower and upper boundaries. As a result, the vertical grid step varies from 10 to 35 km. Because of the spherical geometry, lateral resolution also decreases from 80 km (near the surface) to 40 km (near the CMB). Each cell contains 25 tracers, corresponding to $\approx 1.2$ million tracers in the full model domain, that track mass, composition, and temperature.

### 2.1  Composition

We adopt an idealized mantle composition, consisting of three lithological components: harzburgite (henceforth Hz), basalt (Bs) and primordial material (Prim), as in Gülcher et al. (2021). For simplicity, the symbols Hz, Bs and Prim are henceforth used to indicate the corresponding assemblages at LM conditions. Hz and Bs are defined as solid solutions of two end-member mineral systems: pyroxene-garnet (henceforth *Px-Gt*) and olivine (*Ol*). Accordingly, Hz is 75% *Ol* and 25% *Px-Gt*, while Bs is 100% *Px-Gt*. Prim is not defined based on a specific mineralogical composition, but purely in terms of its physical properties.

Its density profile throughout the mantle is appropriate for *Px-* or bridgmanite-enriched compositions, as described in (Gülcher et al., 2021). Therefore, tracers can be either primordial or represent a mechanical mixture of Bs and Hz.

Each mineral system, as well as Prim, undergoes its own set of solid-to-solid phase transitions, affecting both the density and viscosity profiles of mantle materials. A description of how these effects are parametrized is found in Sect. 2.2. Depths, temperatures, and other parameters for *Ol*, *Px-Gt* and Prim phase transitions are listed in table 1 and are taken from Gülcher et al. (2021) (except those varied as a parameter in this study). The properties of the Prim are instead kept fixed, with values identical to the reference case $\mathbf{M}_{100dD}$ of Gülcher et al. (2021): Prim is 100 times more viscous than pyrolite, and is roughly neutrally buoyant around 1500 km depth. This high intrinsic viscosity is justified by experimental constraints on Bm strength (e.g., Tsujino et al., 2022; Okamoto and Hiraga, 2024), while the density value is consistent with a $(Fe_{0.12}Mg_{0.88})SiO_3$ bridgmanite (Gülcher et al., 2021). Of course, any different choice may lead to a different evolution of the primordial heterogeneities, as described in Gülcher et al. (2020); Gülcher et al. (2021); Ballmer et al. (2017a). However, this particular choice leads to large Prim blobs preserved in the mid-mantle (Gülcher et al., 2020; Gülcher et al., 2021; Ballmer et al., 2017a), allowing us to study how Bs LM properties affect the long-term survival of BEAMS-like heterogeneities.

**Table 1.** List of main parameters used for phase transitions of each system, including Clapeyron slope $\gamma$, density change $\Delta\rho$ and reference bulk modulus $K_0$. The highlighted row indicates the "ghost" transition (see Sect. 2.4). The asterisk denotes parameters that are explored as part of this study, e.g., $\Delta\rho_{Px-Gt}$ ranges from 230 to 430 kg/m$^3$ in steps of 40 kg/m$^3$ and $\Delta\rho_{Ol}$ is computed using Eq. A1.

| Depth [km] | Width of Transition [km] | Temperature [K] | $\gamma$ [MPa/K] | $\Delta\rho$ [kg/m$^3$] | $K_0$ [GPa]; Depth Range [km] |
|---|---|---|---|---|---|
| *Ol*: $\rho_{surf} = 3240$ kg/m$^3$ | | | | | |
| 410 | 25 | 1600 | 2.500 | 180 | 85; 410-660 |
| 660 | 25 | 1900 | -2.500 | 435 | 210; 660-720 |
| **720** | **75** | **1900** | **1.000** | ***** | **210; 720-2740** |
| 2740 | 25 | 2300 | 10.000 | 61 | 210; 2740-2890 |
| *Px-Gt*: $\rho_{surf} = 3080$ kg/m$^3$ | | | | | |
| 40 | 25 | 1000 | 0.000 | 350 | 125; 40-300 |
| 300 | 75 | 1600 | 1.000 | 100 | 85; 300-720 |
| 720 | 75 | 1900 | 1.000 | * | 210; 720-2740 |
| 2740 | 25 | 2300 | 10.000 | 61 | 210; 2740-2890 |
| *Prim*: $\rho_{surf} = 3074$ kg/m$^3$ | | | | | |
| 40 | 25 | 1000 | 0.000 | 260 | 140; 40-380 |
| 380 | 50 | 1600 | 1.675 | 130 | 85; 380-660 |
| 660 | 75 | 1900 | 0.000 | 450 | 230; 660-2740 |
| 2740 | 25 | 2300 | 10.000 | 61 | 210; 2740-2890 |

The composition carried by tracers may be altered over time due to melting. Partial melting of tracers in the Hz-Bs space, for instance, allows to generate the oceanic crust. We approximate melting of Prim tracers by instant conversion of Prim tracers to Bs-Hz tracers with $40\%$ Bs and $60\%$ Hz as soon as they cross the pyroxenite solidus (see Gülcher et al. (2021) and references therein for details).

## 2.2 Density and Rheology

When a system undergoes a phase transition, the corresponding density change $\Delta\rho$ (see table 1) is added. $\Delta\rho$ is one of the variables that are changed as part of the parameter study (see Sect. 2.4). A visco-plastic rheology is applied: materials deform plastically until a critical, pressure-dependent yield stress is reached. Viscous deformation then obeys the Arrhenius law:

$$\eta(P,T) = \eta_0 \lambda exp \left[ \frac{E_a + PV_a}{RT} - \frac{E_a}{RT_0} \right] \tag{1}$$

Where $E_a$, $V_a$, $\eta_0$, $T_0$ and $R$ are, respectively, the activation energy and volume, the reference viscosity and temperature (i.e., at surface conditions) and the ideal gas constant (see table A1 for values). In addition, $\lambda(C, z)$ is a multiplier that parameterizes the viscosity jumps due to phase transitions. Hence, viscosity depends on temperature $T$, composition $C$ and depth $z$. The parameter $\lambda$ is explored as part of our study (see Sect. 2.4).

## 2.3 Initial and Boundary Conditions

The set-up of all our models is identical to the reference model in Gülcher et al. (2021), of which we summarize the main features here. The initial temperature profile is an adiabat with a potential temperature of 1900 K, consistent with a hotter-than-present early Earth (due, e.g., to higher initial concentration of heat-producing elements). Boundary conditions are free-slip and isothermal, with 300 K and 4000 K for surface and CMB temperatures, respectively. All the models in this study are purely bottom-heated, i.e., internal heating is absent. Consistently with several scenarios of MO crystallization (Caracas et al., 2019; Boukaré et al., 2015; Xie et al., 2020), we choose an idealized, two-layered structure for the initial compositional profile. Accordingly, the bottom layer is composed of Prim, plus the addition of $5\%$ pyrolitic noise. Differently from Gülcher et al. (2021), in which the thickness of the Prim layer $D_{prim}$ is varied as part of the study, we choose a fixed value $D_{prim} = 1650$ km (indeed, the amount of long-term Prim preservation does not depend on the initial thickness of the Prim layer, see Gülcher et al., 2021). Finally, the upper layer is initialized as a roughly pyrolitic mechanical mixture ($15\%$ Bs and $85\%$ Hz; henceforth simply referred to as 'pyrolite', or Py).

## 2.4 Parameters Explored

To investigate the influence of ROC properties on mantle long-term evolution and mixing style of heterogeneity, we systematically explore the density and viscosity contrasts of basalt with respect to pyrolite in the lower mantle. These are respectively defined as follows:

$$\frac{\Delta\rho_{Bs}}{\rho_{Py}} = \frac{\rho_{Bs} - \rho_{Py}}{\rho_{Py}} \bigg|_{z=1500 \text{ km}} \tag{2}$$

$$\zeta = \left.\frac{\eta_{Bs}}{\eta_{Py}}\right|_{z=1500 \text{ km}} \tag{3}$$

Where $\rho_{Bs}$ and $\rho_{Py}$, $\eta_{Bs}$ and $\eta_{Py}$ are computed along the reference adiabat with potential temperature of 1600 K. Since both $\frac{\Delta\rho_{Bs}}{\rho_{Py}}$ and $\zeta$ vary in principle across the mantle, we only report their value at the designated depth $z = 1500$ km, when labeling cases. This depth corresponds to the mid-mantle region and is taken here to be roughly representative of the entire LM.

To achieve the desired density and viscosity contrasts, we modulate the reference density jump and viscosity jump of *Px-*
140 *Gt* ($\Delta\rho_{Px-Gt}$ and $\lambda_{Px-Gt}$, respectively) at the 720 km phase change (see table 1). We ensure that the physical properties of pyrolite (i.e., reference density and viscosity profiles) remain fixed as parameters $\frac{\Delta\rho_{Bs}}{\rho_{Py}}$ and $\zeta$ are varied in our study (see Desiderio and Ballmer (2024) and Sect. A). Since the pyrolitic mantle is defined as a Bs-Hz mechanical mixture, every variation imposed on the physical properties of *Px-Gt* (pure Bs) has to be matched by a corresponding (and opposite) change in the properties of *Ol* (which affects Hz). This is illustrated in Fig. 1, displaying the reference density profiles of both Hz and
145 Bs (in the LM).

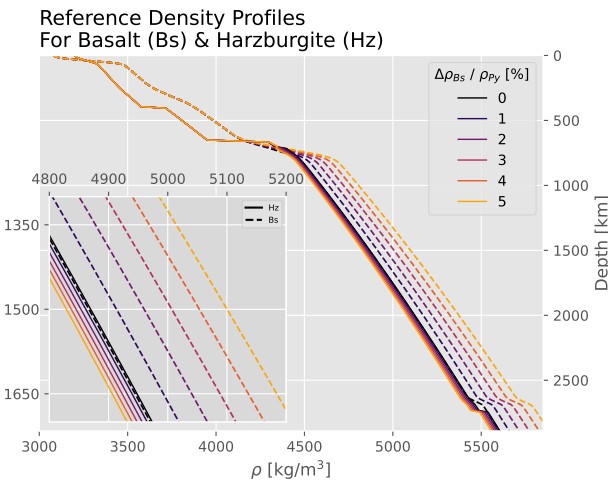

**Figure 1.** Reference density profiles of Hz and Bs used in this study. These profiles are computed along an adiabat with potential temperature $T = 1600$ K. The profile is not shown for pyrolite, for visual clarity.

As the depths at which the *Ol* and *Px-Gt* systems change to their respective lower-mantle phase assemblages do not exactly match ($\sim 660$ and $\sim 720$ km, respectively; see table 1), any correction that is applied to maintain the pyrolite profiles fixed is expected to fail in the [660, 720] km depth range. This would result in a spike in pyrolite density and viscosity with increasing magnitude for increasing $\Delta\rho_{Bs}/\rho_{Py}$ and $\zeta$, respectively. To correct this, an additional "ghost" transition is added for the *Ol*
system at 720 km, where the changes in viscosity and density profiles are applied such that the density and viscosity profiles of pyrolite remain fixed as a function of $\Delta\rho_{Bs}/\rho_{Py}$ and $\zeta$. We stress that this transition has no physical meaning and its only purpose is to avoid any artifacts in the physical properties of pyrolite.

We explore the main parameters in this study, $\frac{\Delta\rho_{Bs}}{\rho_{Py}}$ and $\zeta$, as follows: for the lower-mantle density contrast of Bs, $\frac{\Delta\rho_{Bs}}{\rho_{Py}}$, six different values are tested, ranging from $0\%$ (i.e., Bs is neutrally buoyant in a pyrolitic ambient mantle) to $5\%$, in steps of $1\%$, motivated by the range determined by mineral physics experimental results (Hirose et al., 2005; Ricolleau et al., 2010; Tsuchiya, 2011). For the viscosity contrast $\zeta$, we explore values of 1 (i.e., Bs is intrinsically as strong as pyrolite), 10, and 100. We also test $\zeta = 0.1$ for the cases $\frac{\Delta\rho_{Bs}}{\rho_{Py}} \in [1\%, 3\%]$, motivated by the experimental results of Immoor et al. (2022). The resulting depth-profiles of the density contrasts relative to pyrolite and of $\eta$ are shown in Figs. A1 and A2, respectively, for all relevant mantle materials.

Finally, as a preliminary exploration of primordial material preservation under the effects of internal heating, we perform a secondary suite of four cases, with $\frac{\Delta\rho_{Bs}}{\rho_{Py}}$ and $\zeta$ of $2\%$ and 0.1; $2\%$ and 1; $3\%$ and 0.1; as well as $3\%$ and 100, respectively. For these cases, internal heating is treated using a single effective Heat-Producing Element (HPE) with a half-life $\tau = 2.24$ Gyr (Schubert et al., 2001; Yan et al., 2020). HPEs are not uniformly distributed in the mantle: the initial primordial material is entirely depleted of HPEs – consistent with early MO cumulates (Corgne et al., 2005; Labrosse et al., 2007) – while the initial heat-production rate $H$ in the pyrolitic mantle is increased in order to maintain a chondritic budget for the entire planet (i.e., $H = H_0/(1-r)$, where $H_0 = 15.8 \times 10^{-12}$ W/kg is the initial internal heating rate (Yan et al., 2020), and $r = 0.47$ is the initial mass fraction of primordial material in the mantle). Secondly, shallow mantle melting continuously creates an HPE-enriched crust and leaves a depleted residue, using a solid/melt partition coefficient $D = 0.01$.

## 3    Results

In this section, we first present the results of our main numerical model suite (the four additional cases with HPEs are described separately in Sect. 3.3 and in more detail in Sect. C). We run 21 cases for 5 Gy of model time, with variable viscosity ratios $\zeta$ and density contrasts $\Delta\rho_{Bs}/\rho_{Py}$ between basalt (Bs) and the pyrolite in the lower mantle. The models are initialized with an intrinsically viscous primordial layer at depth $> 1240$ km, while Bs and harzburgite are self-consistently formed over time due to mantle melting. The primordial material is meant to represent ancient Si-enriched material (e.g., bridgmanitic magma-ocean cumulates) with a viscosity 100 times higher than pyrolite and moderate density excess, consistent with a $(Mg_{0.88}Fe_{0.12})SiO_3$ bridgmanite (as in Gülcher et al., 2021, see Sect. 2). Figure 2 shows snapshots of the compositional field after $t = 4.5$ Gy model evolution: a variety of heterogeneity mixing styles is obtained, reflecting a distinct long-term mantle evolution for various cases, although all our cases undergo a similar evolution during the first few hundreds My model time.

### 3.1    Early Model Evolution

Melting-induced differentiation of the pyrolitic uppermost mantle creates a basaltic oceanic crust and a complementary harzburgitic lithosphere (see video supplement). Meanwhile, hot and cold Thermal Boundary Layers (TBLs) develop near the CMB and surface, respectively. Subsequently, sluggish convection develops in the upper mantle, with cold downwellings from the top boundary layer. Cold downwellings eventually pile up atop the Prim-Py compositional interface at 1240 km depth, such that the uppermost pyrolitic lower mantle cools down along with the upper mantle. In contrast, the underlying primordial portion of the

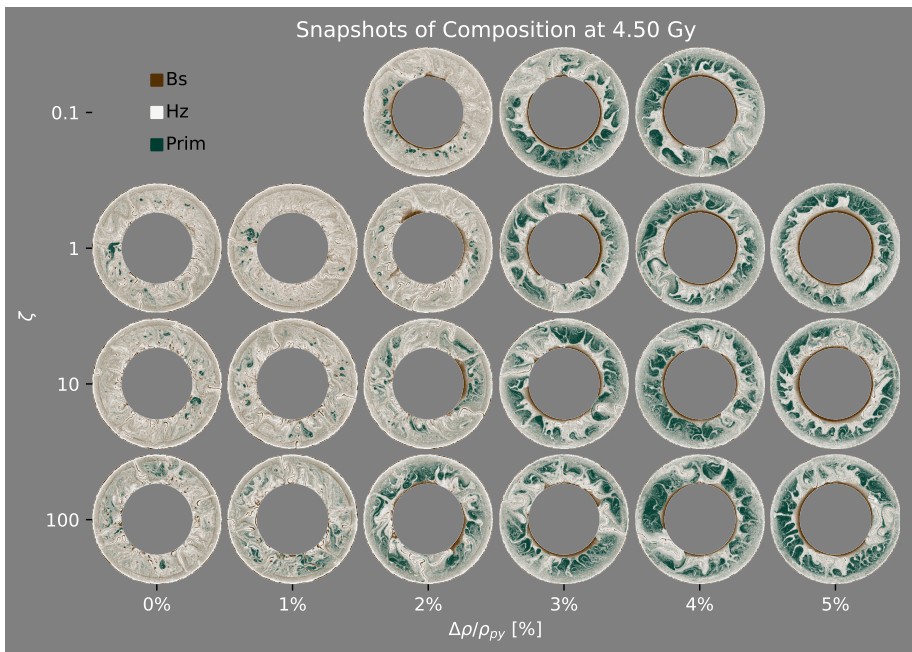

**Figure 2.** Snapshots of composition taken at $t = 4.5$ Gy of model evolution. The legend indicates the three colors corresponding to each discrete end-member composition; the continuous colormap is reproduced in Fig. B1. Cases with different pyrolite-Bs viscosity contrasts, $\zeta$, and density contrasts, $\Delta\rho_{Bs}/\rho_{Py}$, in the lower mantle are organized in different rows and columns, respectively (as labeled).

lower mantle is continuously heated from below. The growing temperature difference across the compositional interface, along with subducted lithosphere piling up on top of it, promotes a gravitational instability that eventually triggers a mantle-scale compositional overturn. During this overturn, the primordial layer is separated into large discrete blobs. Primordial blobs are stabilized in the center of convection cells, separated by up- and down-welling conduits of pyrolitic material, and intermittently rotate as they interact with the surrounding up- and downwelling flow (Gülcher et al., 2020; Gülcher et al., 2021; Ballmer et al.,

2017b; Manga, 1996; Becker et al., 1999). These large primordial blobs are occasionally laterally displaced or separated into smaller blobs (see video supplement). Upwellings originating from the CMB open new conduits between primordial blobs or rise through existing ones, dragging portions of primordial material upwards. Through these mechanisms, some primordial material is passively entrained into the upper mantle. In the upper mantle, where primordial material is assumed to lose its intrinsic strength due to phase transformation (see Sect. 2), it is efficiently mixed, eventually transported to the base of the

lithosphere, and processed by near-surface melting. At this point (i.e., above the pyroxenite solidus), primordial tracers are converted to "regular" tracers in the Hz-Bs space (with Hz-Bs proportions of 40:60; see Sect. 2). Meanwhile, as slabs reach the bottom of the mantle, ROC segregation is facilitated by the phase transition to low-viscosity post-perovskite, heating, and/or small-scale convection (Christensen and Hofmann, 1994; Nakagawa and Tackley, 2011; Yan et al., 2020).

## 3.2 Regimes Overview

ROC physical properties determine the subsequent evolution and style of primordial/recycled heterogeneity preservation. This history of mantle convection and mixing is reflected in the final distribution and amounts of chemical heterogeneities. Accordingly, to quantitatively classify our models into different regimes, we measure the following key variables (averaged between 4 and 5 Gy):

1. mean Bs fraction $f_{Bs}$ in the deep mantle (i.e., averaged in the depth-range $2400 - 2890$ km);

2. mean Prim fraction $f_{Prim}$ in the lower mantle (i.e., in the range $720 - 2400$ km);

3. CMB coverage from Bs, $c$.

The regimes obtained are defined below and are mapped out in Fig. 3, which also shows the key variables introduced above. Using the threshold $f_{Prim} = 0.3$, two main regimes can be established: the first regime (with $f_{Prim} < 0.3$) is characterized by efficient mixing of primordial heterogeneity with the rest of the mantle, while the second regime is characterized by extensive 210 and long-lasting preservation of primordial material in the form of coherent, viscous blobs. Accordingly, we label these regimes "M" and "B", standing for *Mixed* and primordial *Blobs*, respectively.

Each main regime is further subdivided based on the style of ROC preservation obtained alongside primordial material – i.e., using $c$ and $f_{Bs}$. Within regime M in particular (i.e., $f_{Prim} < 0.3$), the criteria $c < 0.6$ and $f_{Bs} < 0.3$ define a sub-regime in which both Bs and Prim are efficiently mixed with the rest of the mantle. Conversely, for $0.6 < c < 0.8$ and $f_{Bs} > 0.4$, while 215 Prim is not efficiently preserved, Bs piles are stabilized above the CMB. We label these sub-regimes "M0" and "MP" – i.e., well-mixed Prim without and with Bs *Piles*, respectively.

Within regime B (i.e. $f_{Prim} \geq 0.3$), the criterion $0.6 < c < 0.8$ defines a sub-regime in which both Bs piles and Prim blobs coexist. High coverage values, i.e. $c > 0.8$ and up to $c \sim 1$, define a sub-regime where a Bs layer is obtained instead of piles. These sub-regimes are accordingly labeled "BP" and "BL" – i.e., Prim *Blobs* with Bs *Piles* and with a Bs *Layer*, respectively.

As shown in Fig. 3, regime B is obtained for high $\Delta \rho_{Bs}/\rho_{Py}$ (and $\zeta$) at the expense of regime M. In particular, M0 and BL are obtained for $\Delta \rho_{Bs}/\rho_{Py} \leq 1\%$ and for $\Delta \rho_{Bs}/\rho_{Py} \geq 3\%$, respectively. Conversely, for intermediate $\Delta \rho_{Bs}/\rho_{Py} = 2\%$, sub-regimes MP and BP correspond to $\zeta \leq 10$ and $\zeta > 10$, respectively (see Fig. 3).

Additionally, Fig. 4 shows depth-profiles for $f_{Bs}$, $f_{Prim}$, $T$, $\eta$, further averaged between 4 and 5 Gy of model time. The panels show the different sub-regimes, with the shaded areas encompassing the area covered by the time-averaged profiles 225 of all models that belong to each sub-regime. The profiles reflect the differences between the sub-regimes described above. Indeed, Fig. 4 shows that M0 and MP are both characterized by low $f_{prim}$ across the entire mantle. Compared to M0, models with piles (i.e., both the MP and BP sub-regimes) show high Bs enrichment in the lowermost mantle, i.e., below $\sim 2600$ km depth). Further, the case in BP is characterized by higher $f_{Prim}$, whose profile peaks at around $z = 1500$ km, also revealing a preferential depth at which blobs are preserved. In sub-regime BL, the deep Bs-rich region is thicker compared to MP and BP, 230 and has also generally higher $f_{Bs}$. Moreover, $f_{Prim}$ is higher as well, and blobs preferentially survive between 1500 and 1000 km depth, as seen in Fig. 4). Moving from regime M towards BP and BL, profiles show that global mantle viscosity increases

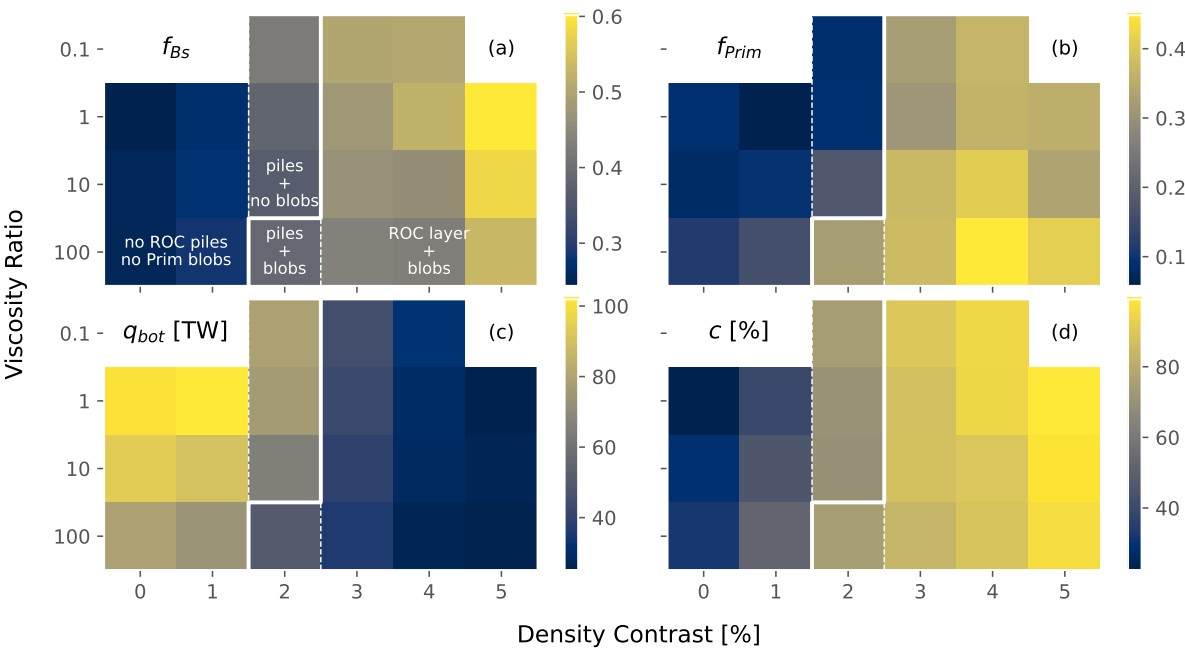

**Figure 3.** Regime diagram spanning the parameter space of our study (i.e., the viscosity ratio $\zeta$ and the density contrast $\Delta\rho_{Bs}/\rho_{Py}$ between ROC in the LM and the pyrolitic ambient mantle). Panels a, b, c and d represent basalt and primordial fractions, CMB heat-flux and CMB basalt coverage ($f_{Bs}$, $f_{Prim}$, $q_{bot}$ and $c$), respectively. All quantities are averaged between 4 Gy and 5 Gy model time. We calculate $f_{Prim}$ and $f_{Bs}$ by averaging across the depth-ranges 720 km$\leq z \leq$ 2400 and 2400 km$\leq z \leq$ 2890 km, respectively. We calculate $c$ as the fraction of cells above the CMB with $f_{Bs} > 0.5$. The boundary between the main regimes 'Mixed Prim' and 'Prim Blobs' is drawn by setting the threshold $f_{Prim} = 0.3$ and is indicated with a full line. The sub-regimes are characterized as having 'No Piles', 'deep ROC Piles' and a 'deep ROC Layer'. Boundaries between these sub-regimes are drawn by setting $c = 0.6$ and $c = 0.8$, and are indicated with a dashed line. Regime labels appear only in panel a to avoid clutter.

in response to both the increase in $f_{Prim}$ (as the primordial material is intrinsically strong) and the slight decrease in average $T$. Finally, as Bs piles and layers are heated over time, a thick, lower thermochemical boundary layer is formed in sub-regimes MP, BP, and BL.

To better understand the sub-regimes introduced above, we describe the evolution of four representative example cases in detail below.

**M0: Well Mixed Primordial Material Without thermochemical Piles.** For $\Delta\rho_{Bs}/\rho_{Py} = 0\%$ and $\zeta = 1$ (see Fig. 2), primordial blobs remain largely coherent for $\sim 2-3$ Gy, but are eventually split into progressively smaller fragments by 240   frequent up- and down-wellings (find corresponding case in the video supplement). In the presence of vigorous whole-mantle

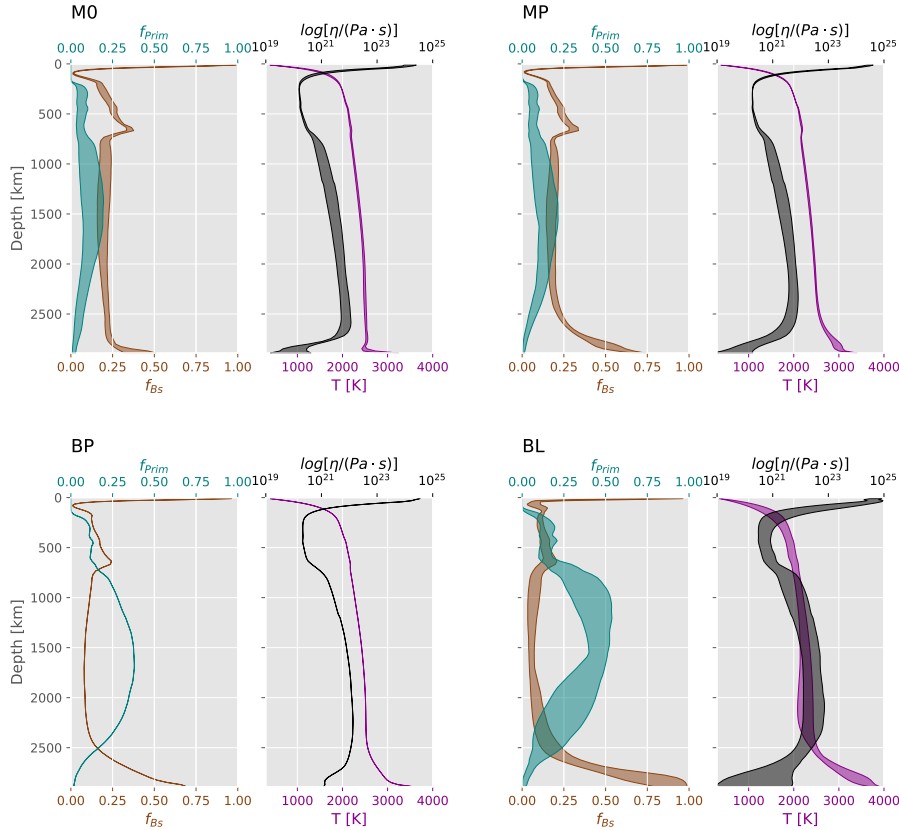

**Figure 4.** Radial profiles of Bs fraction $f_{Bs}$, Prim fraction $f_{Prim}$, Temperature $T$ and Viscosity $\eta$ for each sub-regime. Shaded areas represent the range of variation of the time-averaged profiles of all models within each sub-regime. For regimes M0, MP and BL, respectively, the lower and upper boundaries of the represented $\eta$-profile ranges correspond to: 1%, 1.0 and 1%, 100; 2%, 0.1 and 2%, 10; 3%, 0.1 and 5%, 1 (in terms of $\Delta\rho_{Bs}/\rho_{Py}$ and $\zeta$).

convection, these fragments are continuously and rapidly rotated, and progressively eroded due to entrainment: consequently, after 4.5 Gy, surviving primordial blobs are only few and small. Meanwhile, during the entire model evolution, segregated ROC is continuously delivered to the lower TBL by deep-sinking slabs. However, once segregated from the slab, ROC does not accumulate near the CMB due to its low intrinsic density. Instead, segregated ROC mostly remains suspended within the convective flow to be circulated and mixed throughout the mantle. In the end, the mantle has the appearance of a "marble cake", with a few small primordial "specks" and subducted lithosphere/ROC slabs embedded in a pyrolitic matrix.

**MP: Well Mixed Primordial Material + Thermochemical Basaltic Piles.** For $\Delta\rho_{Bs}/\rho_{Py} = 2\%$ and $\zeta = 1$, primordial blobs are also efficiently eroded over time: at 4.5 Gy, only few small primordial blobs are preserved, similar to the previous case. However, within $\sim 2$ Gy of model evolution, segregated ROC starts to accumulate above the CMB into progressively larger thermochemical piles, due to its moderate (i.e., greater than above) density contrast (see video supplements). These

piles are occasionally displaced laterally by incoming slabs, but not efficiently entrained by convective flow, compared to the rate at which Bs is continuously segregated in the lower TBL. Thus, they survive for the whole duration of the model evolution, although they do not entirely cover the CMB. In the end, the compositional structure of the mantle involves ROC thermochemical piles, and is otherwise similar to the "marble cake" as predicted in the previous case.

**BP: Primordial Blobs + Thermochemical Basaltic Piles.** For $\Delta\rho_{Bs}/\rho_{Py} = 2\%$ and $\zeta = 100$, a shift in the final compositional structure of the mantle is obtained (see Fig. 2). As above, up- and down-wellings fragment and erode primordial blobs, opening pyrolitic conduits between them. However, a stable configuration is eventually achieved, wherein primordial, intrinsically strong blobs tend to remain confined at the center of convection cells, while deformation is mostly focused in the weaker pyrolitic mantle that surrounds them (see video supplements). As up-/down-wellings are mainly accommodated by these weak 260 conduits, the cores of the blobs themselves remain largely undeformed, as they are continuously rotated, but not efficiently eroded. At the same time, ROC segregates and accumulates in the deep mantle efficiently, similar to the previous case. Thus, after 4.5 Gy, large primordial blobs (i.e., $\sim 1000-1500$ km in length-scale) dominate the lower mantle, extending from a depth of 720 km down to 2000-2800 km (see Fig. 2). Slabs are strewn across the pyrolitic space between blobs. Finally, ROC piles are stabilized in the lowermost mantle.

**BL: Primordial Blobs + Thermochemical Basaltic Layer.** For $\Delta\rho_{Bs}/\rho_{Py} = 4\%$ and $\zeta = 1$, primordial blobs are not efficiently eroded, similar to the previous case. Meanwhile, the greater ROC density contrast enhances the segregation of ROC (from harzburgite) in subducted slabs close to the CMB. As a result, Bs accumulation at the base of the mantle starts earlier (at $\sim 1.5$ Gy) and is more efficient compared to all previous cases. Within $\sim 2$ Gy, thermochemical piles grow and join into a laterally mostly homogeneous thermochemical layer, covering most of the CMB (see video supplements). Slabs may 270 occasionally displace the layer and momentarily clear regional patches of the CMB (i.e., holes in the layer). In the end, the mantle compositional structure is dominated by large primordial blobs with pyrolitic conduits (which itself consists of a marble cake of harzburgite and ROC streaks) between them, and a lowermost mantle highly enriched in Bs.

### 3.3   Parameter Sensitivity

Beyond the regime classification, predicted material distributions systematically depend on model parameters. The histograms 275 of compositions in Figs. 5 and 6 convey how much and in what form Bs and Prim are preserved in the deep and lower mantle, respectively. The histograms in Fig. 5 count the Bs fractions averaged in depth-columns below $z = 2400$ km, denoted with $\bar{f}_{Bs}$. The histograms in Fig. 6 count the Prim fractions averaged in depth-columns between 720 and 2500 km, denoted with $\bar{f}_{Prim}$ (we further define $P_{0.25}$ as the percentage of depth-columns characterized by $\bar{f}_{Prim} \geq 0.25$ – also indicated in the figure).

    In all cases of sub-regime M0, $\bar{f}_{Bs}$ histograms are narrowly distributed around the pyrolitic value of $\sim 0.2$. As $\Delta\rho_{Bs}/\rho_{Py}$ 280 is increased, ROC segregation efficiency and gravitational stability are enhanced, driving the accumulation of basaltic material at the CMB. Accordingly, histograms for the piles and layer sub-regimes become bimodal (with a second mode at $\bar{f}_{Bs} \sim 0.5-0.6$), which are related to the presence of enriched ROC piles and the pyrolitic regions in-between. For extreme density values, the balance between ROC entrainment and segregation is strongly skewed towards the latter: thus, the CMB is (almost always) covered by ROC at $t = 4.5 \pm 0.5$ Gy (see Figs. 2, 3, 5).

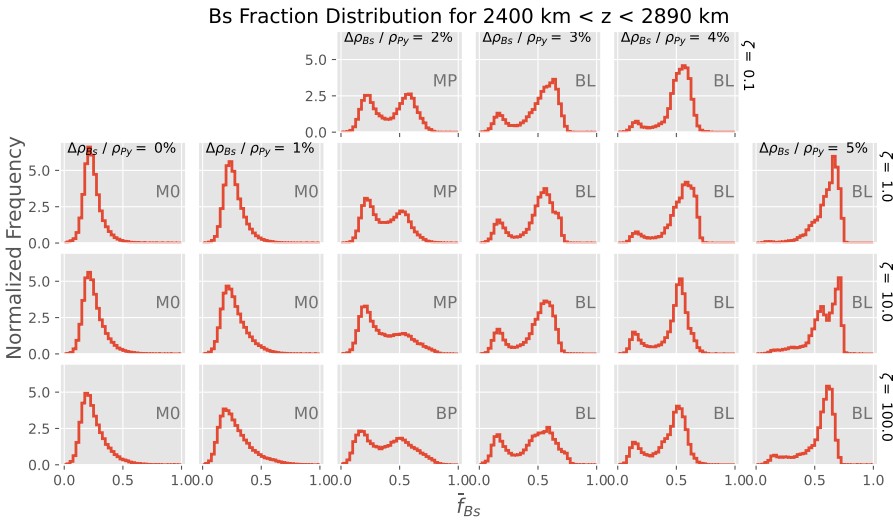

**Figure 5.** Histograms of Bs fraction $f_{Bs}$ are constructed by averaging it in the columns below $z = 2400$ km, in the range $t = 4.5 \pm 0.5$ Gy. Each histogram is normalized such that the total area is one. Panels are ordered according to increasing $\Delta\rho_{Bs}/\rho_{Py}$ from left to right and increasing $\zeta$ from top to bottom.

The style of ROC accumulation is instead only marginally affected by changes in $\zeta$. In sub-regime M0, increasing $\zeta$ somewhat enhances $c$, as high-viscosity Bs is less efficiently entrained (Gülcher et al., 2021; Manga, 1996; Becker et al., 1999; Davaille et al., 2002; Heyn et al., 2018): however, this alone cannot lead to pile-formation, since low-density, segregated Bs quickly becomes positively buoyant upon heating regardless of $\zeta$ (e.g., see $f_{Bs}$ in the lowermost mantle in Fig. 3). At higher $\Delta\rho_{Bs}/\rho_{Py}$ values, $\zeta$ does not significantly affect ROC accumulation, and, thus, pile/layer coverage (see Fig. 3). This result is also reflected in the Bs distributions (see Fig. 5). Further, higher Bs intrinsic viscosities reduce the convective vigour within the piles (or layer), which thus become more stagnant and stratified (pyrolitic at the top and Bs-enriched at the bottom) as $\zeta$ is increased (Desiderio and Ballmer, 2024). Weak ROC piles (i.e., for $\zeta \leq 0.1$) are instead internally convecting and, thus, more homogeneous (Desiderio and Ballmer, 2024). Thus, average Bs fractions decrease with $\zeta$, while pile/layer coverage is largely unaffected (see Fig. 3).

The preservation of primordial material is enhanced by $\Delta\rho_{Bs}/\rho_{Py}$, as ROC accumulation above the CMB, beginning early in the model history (see video supplement), affects the long-term thermal evolution of the mantle. Indeed, the bottom heat-flux $q_{bot}$ decreases with increasing $\Delta\rho_{Bs}/\rho_{Py}$ (see Figs. 3 and C6), as even partial ROC accumulation above the CMB insulates the mantle from the hot core, and diminishes the magnitude of conductive heat flow (Desiderio and Ballmer, 2024; Citron et al., 2020; Mulyukova et al., 2015; Panton et al., 2023). The prolonged reduction in heat carried upwards over time also leads to a lower average mantle temperature and a higher mantle viscosity (see Fig. 4). Indeed, the viscosity profiles of sub-regime M0 are comparable to those of sub-regime MP (see Fig. 4); conversely, the viscosity profile for BP is comparable to the lower end

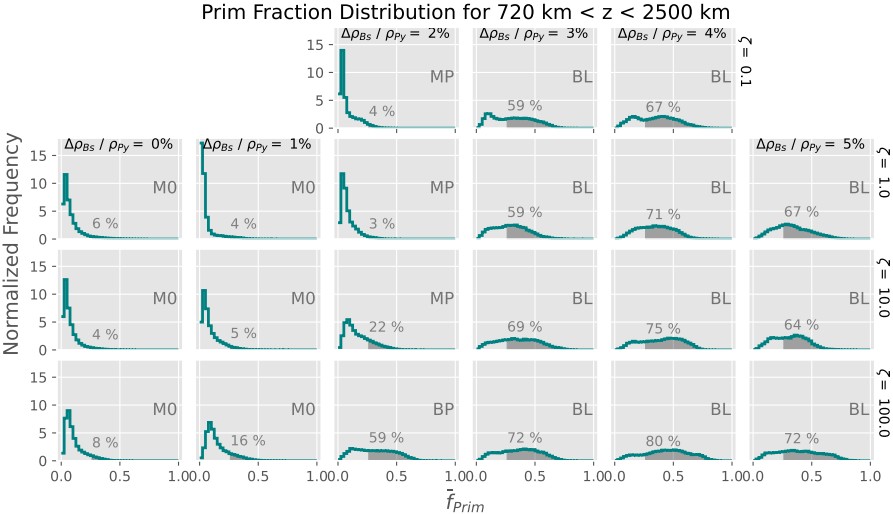

**Figure 6.** Histograms of Prim fraction $\bar{f}_{Prim}$ averaged in depth-columns between 720 and 2500 km, in $t = 4.5 \pm 0.5$ Gy. Histograms are normalized such that their total area is one. The area for which $\bar{f}_{Prim} \geq 0.25$ is highlighted, along with its value in percentage. Panels are ordered according to increasing $\Delta\rho_{Bs}/\rho_{Py}$ from left to right and increasing $\zeta$ from top to bottom. Sub-regimes are also labeled.

of the range shown for BL viscosities. As the vigour of global-scale mantle convection is decreased, the mixing efficiency of primordial heterogeneity is also diminished (see Fig. 6).

Further, as the hot core is shielded by accumulated ROC, upwellings from clear areas of the CMB are hotter than the ones
originating from the top of recycled domains (Gülcher et al., 2021). Thus, ROC accumulation reduces primordial mixing also by decreasing plume vigour – especially in sub-regime BL, due to a more homogeneous CMB coverage. This result is similar to that described by Gülcher et al. (2021) for a dense, primordial lower layer.

We also find that the preservation of primordial material is generally enhanced by $\zeta$. This result is best explained by first focusing on the MP-BP sub-regimes transition, noting that high-viscosity ROC promotes stagnation and stratification within
piles. Within MP, internally convecting and weakly stratified piles (i.e., $\zeta < 10$) allow for more efficient heating and mixing of the mantle (Desiderio and Ballmer, 2024), leading to efficient erosion of primordial blobs within 4.5 Gy (see, e.g., Fig. 3). As $\zeta$ is increased, a thicker conductive thermochemical boundary layer is developed near the CMB as a result of pile stagnation (Desiderio and Ballmer, 2024): thus, $q_{bot}$ is decreased, reducing global convective vigour and mixing of primordial blobs (see, e.g., Fig. 3). Finally, sub-regime BP is characterized by strongly stratified and more effectively insulating thermochemical
piles, leading to even larger Prim blobs (see Figs. 2, 3, 6). In sub-regime BL, efficient basalt accumulation due to high intrinsic density leads to a global, thick thermochemical boundary layer that is even more efficiently insulating. Thus, although $\zeta$ may modulate convective vigour in the basal layer, a higher value only slightly increases $f_{Prim}$ (see Figs. 3 and 6).

These effects are quantified in Fig. 7, showing that increasing $\zeta$ reduces convective vigour in the ROC piles/layer (also see video supplement), and that $f_{Prim}$ is enhanced as a result. Convective vigour in the ROC-rich domains is estimated by

computing the standard deviation $s_{u_z}$ of vertical velocity $u_z$ between 4 Gy and 5 Gy (a model cell is considered as part of a pile/layer if the corresponding $f_{Bs} > 0.5$ and if $z \geq 2400$ km). Symbols are coloured according to $f_{Prim}$ (as calculated in Sect. 3.2) to show the effect of convection across the pile/layer on Prim preservation. Convective vigour in the Bs piles and Bs layer is further quantified in Fig. B2, showing raw 2D histograms of Bs fraction and vertical velocity $u_z$ in the interval $t = 4.5 \pm 0.5$ Gy. Additionally, Fig. B3, shows temperature profiles at $t = 4.5$ Gy taken at different azimuths across Bs piles (and similarly

across Bs layers in Fig. B4). Each figure compares two cases with low and high intrinsic Bs viscosity: consistent with internal convection, temperature profiles are roughly adiabatic within the piles/layers when Bs viscosity is low; for high Bs viscosity, temperature profiles show a single thick TBL across the stratified piles/layers instead, highlighting conduction as the dominant form of heat transport (Desiderio and Ballmer, 2024).

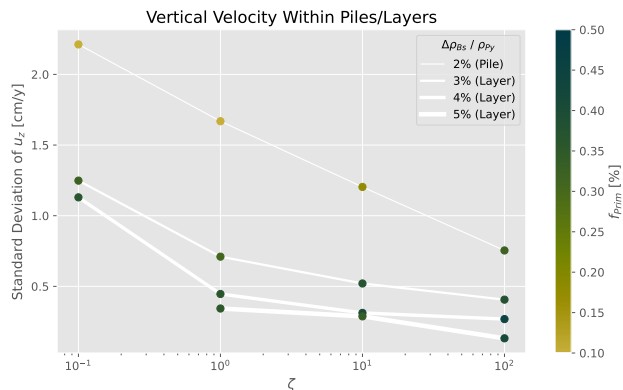

**Figure 7.** Standard deviation of vertical velocity $s_{u_z}$ within cells with Bs fraction $f_{Bs} > 0.5$, in the lowermost $\approx 400$ km of the mantle and in the time-range $t = 4.5 \pm 0.5$ Gy, for $\Delta\rho_{Bs}/\rho_{Py}$ from 2% to 5% (we exclude cases not characterized by piles – i.e., sub-regime M0). Data points are colored according to the corresponding $f_{Prim}$, as calculated in Sect. 3.2.

Bs intrinsic viscosity can exert an effect on mantle thermal evolution only if Bs accumulates above the CMB in some form:

accordingly, $\zeta$ has marginal control on Prim preservation in sub-regime M0 (which is thus not represented in Fig. 7). However, within M0, we note that increasing $\zeta$ to extreme values increases the size and number of surviving primordial blobs (see Figs. 2, 3, 6). Indeed, as delayed mixing of strong Bs slightly enhances enrichment of high-viscosity ROC in the lower TBL, plume vigour may be dampened (Kumagai et al., 2008; Lin and van Keken, 2006), leading to less efficient erosion of primordial material.

Furthermore, we highlight the possible feedback between the preservation of primordial and recycled heterogeneities: as ROC enrichment increases in response to $\Delta\rho_{Bs}/\rho_{Py}$, the rate of erosion of primordial material is reduced. This leads to a growth in the Prim blobs' size, which may mechanically shield the recycled domains from downwellings. Recycled domains are thus less likely to be displaced, favoring their persistence over time and enhancing their protective action on the primordial blobs themselves. Such a mechanism for mutual Prim-ROC preservation has also been suggested by (Gülcher et al., 2021).

Finally, to explore the effects of radiogenic heating on primordial-material preservation, we run four additional cases, for which composition-dependent internal heating is taken into account (see Sect. 2.4). We test four cases, with $\frac{\Delta\rho_{Bs}}{\rho_{Py}}$ and $\zeta$ of 2% and 0.1; 2% and 1; 3% and 0.1; as well as 3% and 100, respectively (the first two parameter combinations are drawn from sub-regime MP of the main suite, while the other two would correspond to the BL sub-regime). We summarize the main results of this secondary suite below, with more details given in Appendix C.

The addition of internal heating slightly changes the early model evolution before the global overturn of the primordial layer, as well as timing of the overturn, but, crucially, does not prevent the overturn itself. In the following, the most important difference between cases with and without HPEs is that, for the former, the mantle is consistently hotter and less viscous. In a hotter mantle with HPEs, Bs accumulation is consistently more efficient compared to the equivalent cases without HPEs, because lower mantle viscosities generally enhance segregation efficiency (Yan et al., 2020). Accordingly, a Bs layer is stabilized at the

CMB, even for $\frac{\Delta\rho_{Bs}}{\rho_{Py}} = 2\%$ (see video supplement and Fig. C1). This Bs layer is enriched in HPEs due to the low solid/melt partition coefficient used for HPEs. Nevertheless, intermediate/high Bs intrinsic viscosities can still decrease convective vigour within the layer (see video supplement). In any case, higher Bs densities still promote the preservation of primordial blobs, also in models with internal heating (see Fig. C1). Perhaps even more importantly, internal heating generally increases the vigour of convection in the ambient pyrolitic mantle surrounding primordial blobs, which leads to enhanced mixing of primordial

material (compared to cases without HPEs). We ascribe this higher efficiency of Prim erosion primarily to the indirect effect of HPEs on mantle temperature/viscosity rather than the direct effect of HPEs on convective style. Thus, internal heating promotes yet another regime besides those obtained with the same parameters but no HPEs (i.e., MP and BL): a Bs layer is always stabilized at the CMB, while primordial blobs are more efficiently mixed – even though average $f_{Prim}$ still ranges between 0.05 and 0.21, for lower and higher $\frac{\Delta\rho_{Bs}}{\rho_{Py}}$, respectively (see Fig. C4). In any case, convection in these models (including those

with no HPEs) is likely more vigorous than what is expected for Earth, as the mantle temperatures obtained here are higher than realistic – even for the early Earth (see discussion in Sect. 4.5). In summary, accounting for the effects of HPEs changes mantle evolution and processing efficiency, but still allows for the long-term preservation of substantial fractions of primordial material (see Sect. C for details).

## 4   Discussion

### 4.1   Results summary and Relation to Previous Work

In a suite of 2D mantle-convection models, we investigated how ROC intrinsic density and intrinsic viscosity in the lower mantle affect the long-term preservation of the primordial material. The models were initialized as a layered compositional structure, with a lower layer of "primordial" material that is 100-fold more viscous than the pyrolitic layer above it, motivated by a range of processes that may have operated in the early Earth (including, e.g., MO crystallization) and which may have

enriched the lowermost mantle in viscous Bridgmanite (Bm) (e.g., Boukaré et al., 2025; Caracas et al., 2019; Nabiei et al., 2021). A range of regimes of mantle mixing is obtained depending on ROC properties explored: for low intrinsic ROC density, both recycled and primordial heterogeneities are efficiently mixed (sub-regime M0; Fig. 2), while higher values promote ROC

accumulation above the CMB, consistent with previous work (e.g., Brandenburg and Van Keken, 2007; Citron et al., 2020; Davaille, 1999; Mulyukova et al., 2015; Nakagawa et al., 2010; Jones et al., 2020). This ROC accumulation is manifested as either isolated piles ($\Delta\rho_{Bs}/\rho_{Py} = 2\%$, sub-regimes MP, BP) or a global deep layer ($\Delta\rho_{Bs}/\rho_{Py} \geq 3\%$, sub-regime BL). ROC accumulations reduce CMB heat flux, global mantle temperature and, thus, convective vigour and mixing efficiency (e.g., Li and McNamara, 2018; Citron et al., 2020; Panton et al., 2023; Mulyukova et al., 2015). Conversely, ROC intrinsic viscosity plays a secondary role in driving ROC accumulation (as found in Desiderio and Ballmer, 2024).

Importantly, large-scale, mid-mantle primordial heterogeneities (i.e., BEAMS) are only preserved if ROC piles (or a global layer) above the CMB are also stabilized, speaking to the close relationship between ROC and primordial material, and their mutual stabilizing effect (see also Gülcher et al., 2021). As Gülcher et al. (2021) have a fixed ROC density of $\Delta\rho_{Bs}/\rho_{Py} = 3\%$, an analogous of our sub-regime M0 is absent from their results.

We further show for the first time that for intermediate ROC density (i.e., $\Delta\rho_{Bs}/\rho_{Py} = 2\%$), mixing of primordial heterogeneity is highly sensitive to ROC intrinsic viscosity (or, equivalently, its viscosity contrast compared to pyrolite, $\zeta$): high-viscosity ROC reduces convective vigour within the piles, thereby reducing heat flux across the lowermost mantle and, ultimately, mantle global temperature (see also Desiderio and Ballmer, 2024). In turn, this insulation of entire sectors of the lower mantle from below reduces the mixing efficiency of the primordial material. Accordingly, BEAMS with significant size are only preserved for $\zeta \geq 10$ (particularly $\zeta = 100$), and preferentially survive directly above ROC piles (Gülcher et al., 2021). These results are valid as long as piles can be stabilized at the CMB. Our cases in the pile regime may either reflect realistic lower-mantle ROC density values (i.e., $\Delta\rho_{Bs}/\rho_{Py} \sim 2\%$ – see, e.g., Ricolleau et al., 2010); or, if ROC is not dense enough to efficiently accumulate (i.e., $\Delta\rho_{Bs}/\rho_{Py} \sim 1\%$ – see, e.g., Stixrude and Lithgow-Bertelloni, 2024), piles may be formed through alternative processes not explicitly modelled in this study (e.g., delayed entrainment of basal-magma-ocean cumulates: see Sect. 4.2).

In a secondary suite of models (see Sects. 2.4 and C), the effect of HPEs on BEAMS survival are tested by accounting for internal heat production in pyrolitic and recycled materials (e.g., Labrosse et al., 2007). Cases with HPEs have a hotter and less viscous mantle than cases without: for all these cases, a ROC-layer thus forms at the CMB, due to enhanced segregation (Yan et al., 2020), even with low-density ROC ($\Delta\rho_{Bs}/\rho_{Py} = 2\%$). Further, BEAMS erosion is enhanced mainly due to vigorous ambient-mantle stirring – a different mechanism than the one described by Becker et al. (1999). Crucially, density still enhances BEAMS preservation, such that moderate fractions of primordial material can be still preserved with $\Delta\rho_{Bs}/\rho_{Py} = 3\%$ (see Fig. C1).

Primordial material preservation also depends on its intrinsic physical properties, which are fixed in our study (kept identical to the reference case of Gülcher et al., 2021). Gülcher et al. (2021) show that the preservation of primordial material as large BEAMS is efficient for a rather wide range of intrinsic densities and viscosities ). For very low intrinsic density anomalies, however, primordial blobs become positively buoyant, as they warm up over time, promoting erosion of BEAMS, and ultimately, transport into the upper mantle. Similarly, low intrinsic viscosity anomalies of primordial material promote entrainment, leading to less efficient preservation (Gülcher et al., 2021). Our work shows that intrinsically dense (and viscous) ROC can trade off with these effects by promoting preservation of primordial blobs via the stabilization of insulating thermochemical piles.

If instead primordial material is both highly viscous and dense, it forms thermochemical piles interspersed with ROC heterogeneity (Gülcher et al., 2021). In this case, varying degrees of mixing between primordial piles and ROC may arise, depending on ROC physical properties, with implications for the geochemical signature of hotspot lavas sampling the deep mantle (Li et al., 2014a). Conversely, relatively dense but low-viscosity primordial material would be efficiently eroded, forming 'diffuse domains' in the lower mantle instead of BEAMS-like domains (Gülcher et al., 2021). We expect both ROC intrinsic density and viscosity to counteract this and promote BEAMS preservation.

Our assumed primordial-to-pyrolite viscosity ratio of 100 is realistic for a rock composed mostly of bridgmanite (Bm) (Tsujino et al., 2022; Okamoto and Hiraga, 2024), as even small-to-moderate amounts (10%-20%) of ferropericlase (Fp) can significantly lower viscosity (Girard et al., 2016; Thielmann et al., 2020; Yamazaki and Karato, 2001). Moreover, enhanced grain growth in Bm would lead to higher BEAMS viscosity in the diffusion creep regime (Fei et al., 2023), especially in long-lived, relatively warm and Bm-rich BEAMS (Gülcher et al., 2021). As grain-growth is accelerated by higher temperatures (Solomatov, 1996), hot, long-lived piles, such as those obtained in this study, may be characterized by large grains (Schierjott et al., 2020), which would further enhance piles stabilization and, indirectly, BEAMS survival. The attenuation model by Talavera-Soza et al. (2025) supports large grain sizes in LLSVPs and is consistent with coarse-grained BEAMS, as long as the latter are located directly above and close to LLSVPs (as predicted by our study and Gülcher et al., 2021). On the other hand, a mid-mantle, high-attenuation layer may challenge this view (Sun et al., 2025).

## 4.2 Early-Earth Differentiation

A range of processes during Earth formation and early differentiation may result in an initial Bm-rich layer in the lower(most) mantle. For example, equilibration of the proto-Earth mantle with that of the Moon-forming impactor (Deng et al., 2019) may lead to layered mantle with a Si-enriched lower layer mostly composed of proto-Earth materials. Alternatively, chemical fractionation during the crystallization of the early magma ocean and/or the basal magma ocean (BMO) may also lead to chemically distinct domains in the mantle. For a range of plausible starting compositions (i.e., with a Mg/Si similar to pyrolite, e.g., Murakami et al., 2024), the liquidus phase at lower mantle conditions is Bm. This implies that mostly bridgmanitic crystals are formed until $\sim 60\%$ solidification of the magma ocean (Caracas et al., 2019; Nabiei et al., 2021; Miyazaki and Korenaga, 2019).

Strong Bm fractionation leading to a bridgmanitic layer requires efficient solid-melt separation. This depends on the style of crystallization. For fractional crystallization, crystals may segregate from the residual melt via gravitational setting, forming a thick layer with a high Mg/Si (Boukaré et al., 2015). In turn, equilibrium crystallization (Solomatov, 2015) would at first lead to a poorly fractionated layer, but the partially liquid mush should eventually compact into a BMO and an overlying bridgmanitic layer (Caracas et al., 2019). Alternative scenarios for BMO formation involve inside-out crystallization of the primary magma ocean due to a solid-liquid density crossover in the deep mantle (Caracas et al., 2019; Labrosse et al., 2007; Mosenfelder et al., 2009), and overturn of Fe-enriched upper-mantle cumulates (Ballmer et al., 2017b; Elkins-Tanton, 2008).

BMO crystallization can also lead to the formation of a deep bridgmanitic layer. As the melt cools, Mg-rich Bm crystallizes (Boukaré et al., 2015; Miyazaki and Korenaga, 2019; Caracas et al., 2019) and is extracted due to fractional crystallization

(Labrosse et al., 2007). As crystallization progresses, cumulates hence incorporate increasing amounts of Fe, and, depending on the initial composition of the BMO, also eventually involve Fp in addition to Bm (Boukaré et al., 2015; Miyazaki and Korenaga, 2019; Caracas et al., 2019; Nabiei et al., 2021). Such a crystallization sequence results in intrinsically dense late-stage cumulates, which have been proposed as the origin of thermochemical piles and may account for the presence of LLSVPs (Labrosse et al., 2007; Boukaré et al., 2025).

Further, the onset of Fp solidification in the BMO cumulate sequence may be delayed for Si-rich BMO starting compositions, e.g. in the overturn scenario (Ballmer et al., 2017b; Elkins-Tanton, 2008). This would lead to a thick Si-rich layer from BMO crystallization alone (Boukaré et al., 2015). Alternative scenarios include delivery of additional Si during BMO crystallization, for example due to exsolution of $SiO_2$ from the cooling core (Hirose et al., 2017), or via recycling of Hadean/Archean crust (Tolstikhin and Hofmann, 2005; Moore and Webb, 2013; Johnson et al., 2014). For these Si-enriched scenarios for the BMO, highly-viscous St may co-crystallize with Bm in the later part of the crystallization sequence (Boukaré et al., 2015).

## 4.3 Geochemical Implications

Long-term preservation of distinct chemical reservoirs in the mantle is key to reconcile the diverse isotopic signatures manifested in the geochemical record (e.g., White, 2015; Hofmann, 1997). OIB and large-igneous-province geochemical anomalies, particularly in terms of short-lived radionuclides and noble gases, point to the preservation of primordial mantle heterogeneity (e.g., Rizo et al., 2016; Mundl et al., 2017; Jackson et al., 2017; Mukhopadhyay, 2012; Caracausi et al., 2016; Porcelli and Elliott, 2008). However, the location and size of these primordial reservoirs in the mantle remains poorly constrained (White, 2015). At the same time, the geochemical signature of OIBs testifies to the recycling of subducted sediments and oceanic crust, as well as their long-term preservation, possibly in the lowermost mantle (Christensen and Hofmann, 1994; Allègre and Turcotte, 1986; Brandenburg and Van Keken, 2007; Delavault et al., 2016). This 'dual' character of the geochemical record is consistent with our models, in which ROC-rich reservoirs are preserved alongside primordial BEAMS (Gülcher et al., 2021).

Our results support this interpretation by showing that both BEAMS and LLSVP/pile materials are slowly, but continuously entrained by the convecting mantle, even when BEAMS are ultimately efficiently preserved as large-scale, discrete domains. This 'diffuse' ambient primordial material is ultimately entrained by plumes into the upper mantle, and processed by hotspot melting, as explicitly predicted by our models. As the erosion and entrainment of primordial material is a two-stage process, primordial material also enters the hot cores of plumes, which sustain hotspot melting (Farnetani and Richards, 1995), in agreement with their primitive isotopic signatures. Particularly, BEAMS are consistent with primitive noble-gas signatures in the OIB record (Mukhopadhyay, 2012; Tucker and Mukhopadhyay, 2014; Porcelli and Elliott, 2008; Starkey et al., 2009), as long as these elements are not highly incompatible at lower-mantle conditions (Coltice et al., 2011). This clashes with the customary assumption of high noble-gas incompatibility in the lower mantle: however, measurements of partition coefficients and/or solubilities at high pressures are sparse (Karato, 2016; Moreira, 2013) and results contradictory (see, e.g., Shcheka and Keppler, 2012; Jackson et al., 2021). Indeed, the most undegassed mantle source reservoir shows evidence of long-term depletion in incompatible trace elements (White, 2015; Zindler and Hart, 1986) – consistent with BEAMS as early (B)MO cumulates (Ballmer et al., 2017a). Also, theoretical work suggests that noble gases may become more compatible at higher

pressures (Stixrude et al., 2009). On the other hand, if noble gases indeed remain highly incompatible at high pressures, melt trapping in (B)MO cumulates (Jackson et al., 2021) may reconcile BEAMS as primordial geochemical reservoirs. Overall, better constraints on noble-gas partitioning coefficients are needed to test the hypothesis of BEAMS as reservoirs of such primordial signatures, especially at pressures/temperatures relevant for (B)MO crystallization (Moreira, 2013).

The preservation of Si-rich (basal) magma-ocean cumulates may reconcile major-element constraints for Earth composition. In particular, such a primordial preservation would shift the supra-chondritic Mg/Si of the accessible Earth mantle (Palme and O'Neill, 2014) closer to the proposed range of solar-chondritic values (i.e., Mg/Si of about 1∼1.1, see e.g., Murakami et al., 2024). On the other hand, the Refractory Litophile Elements (RLEs) budget of the accessible mantle poses an upper limit of $\sim 13$ wt% on the admissible portion of the mantle occupied by predominantly bridgmanitic domains (Liebske et al., 2005)
– even though this estimated limit relies on partition coefficients measured at 25 GPa, while Bm fractionation in the (basal) magma-ocean would have occurred at higher pressures. Indeed, for a case in the pile regime with intermediate-size BEAMS (e.g., $\zeta = 10$), primordial material accounts for $\sim 13\%$ of the total mass of the modeled mantle. The coupled preservation of BEAMS and partly-ancient thermochemical piles (with a significant fraction of $\geq 2.0$ Ga ROC), as predicted by our study, can not only reconcile the Mg/Si ratio, but also the Ca/Al ratio of the convecting mantle with solar-chondritic values (Murakami
et al., 2024).

## 4.4  Geophysical Signatures

BEAMS have yet to be clearly detected by seismic tomography. However, dampening and distortion of anomalies due to the tomographic inversion process may limit attempts to identify these heterogeneities in tomographic images of the lower mantle (Ritsema et al., 2007). We anticipate that BEAMS detection is inherently challenging for seismic tomography. Our models show
that BEAMS are characterized by only a weak temperature excess compared to the ambient mantle, which would result in a low-amplitude (negative) seismic wavespeed anomaly. This thermal effect would trade off with that of composition (Ballmer et al., 2017a), as Bm is seismically faster than pyrolite at lower-mantle conditions. Hence, the resulting seismic velocity anomaly of BEAMS may be too weak to be detected by seismic imaging methods (Gülcher et al., 2021; Houser et al., 2020). Quantifications of BEAMS-like seismic wavespeed anomalies, however, remain unexplored. Further, thermoelastic properties
of lower mantle minerals are also not yet fully constrained (Marquardt and Thomson, 2020).

Apart from direct tomographic imaging, BEAMS are consistent with several geophysical observations. First, the underlying physical mechanism for an increase in mantle viscosity as revealed by geoid inversion at $\sim 1000$ km depth remains unexplained (Rudolph et al., 2015). This viscosity increase or "jump" has been linked to slab stagnation and plumes deflection in tomographic images at that depth (Fukao and Obayashi, 2013; French and Romanowicz, 2015; Shephard et al., 2017).
Whereas there is no mineralogical phase change that could account for this increase, lateral compositional changes due to BEAMS (Ballmer et al., 2017a) are an attractive scenario, also consistent with the observation that only a subset of slabs stagnates in the mid-mantle while other sink deeper towards the CMB (e.g., Fukao and Obayashi, 2013).

Fossil slabs that stagnate above bridgmanitic blobs may further account for seismic reflections in the uppermost lower mantle. Such reflections are widespread, and are also observed well away from both LLSVPs and subduction zones (Waszek

et al., 2018; Saki et al., 2022; Vinnik et al., 2010; An et al., 2007; Courtier and Revenaugh, 2008). One such region with a widespread reflector at about 1000 km depth is in the North Pacific (Waszek et al., 2018; An et al., 2007; Yuan et al., 2021; Zhang et al., 2023). Furthermore, a dome-shaped reflector in the South-Pacific uppermost lower mantle (Waszek et al., 2018) is difficult to reconcile with the vertical extension of LLSVPs, especially in light of recent constraints that 'deflate' the LLSVPs to a thickness of $\sim 900$km (Richards et al., 2023) or less (Davaille and Romanowicz, 2020). Alternatively, these seismic reflectors may also be caused directly by the "ceilings" of the BEAMS themselves and/or Si-enriched phases heterogeneities within these domains (Xu et al., 2017).

It should be noted that alternative mechanisms may explain a mid-mantle viscosity increase, for example fabric development in high strain regions around sinking slabs and/or the iron spin transition within Fp (Deng and Lee, 2017; Marquardt and Miyagi, 2015). Interestingly, Shephard et al. (2021); Houser et al. (2020) note that the global signature of this spin transition is absent in 1D Earth seismic models (e.g., PREM, Dziewonski and Anderson, 1981), which would be expected for a well mixed, pyrolitic mantle. The suppression of the seismological footprint of the iron-spin transition in Fp would be instead consistent with pervasive Bm-enriched domains in the lower mantle (Shephard et al., 2021; Houser et al., 2020), although different assumptions in modeling this transition may lead to the opposite conclusion (Trautner et al., 2023).

To further constrain the compositional structure of the lower mantle, a systematic forward modeling of the BEAMS geophysical signature is necessary, for example by mapping the predictions of 2- and/or 3D geodynamic models to seismic velocities (Stixrude and Lithgow-Bertelloni, 2011; Connolly and Petrini, 2002). Higher resolution in seismic tomography models is also needed, although reliably relating seismic velocity variations to temperature and composition remains an unresolved challenge (Marquardt and Thomson, 2020; Schouten et al., 2024; Dannberg et al., 2017).

## 4.5  Future work

Several scientific avenues remain highly relevant as follow-up to our work. For example, our four additional cases with internal heating allow to identify intriguing trends in terms of BEAMS preservation, but warrant further investigation. In particular, while HPEs increase mantle temperatures, thus enhancing convective vigour and BEAMS erosion, high ROC densities still allow for the preservation of (although smaller than without internal heating) BEAMS: HPEs may thus shift the regime boundary between BEAMS entrainment and survival towards higher densities and/or viscosities for ROC and/or primordial material. These potential trade-offs should be investigated in future convection models in Earth-like conditions, since all our additional cases with HPEs display a thick ROC layer – unlike LLSVPs (Tackley, 2002): lower intrinsic density anomalies, thermal and compositional effects on ROC thermal conductivity (Guerrero et al., 2024) and/or lower rates of ROC production and subduction (Panton et al., 2023) may restore these cases to the pile regime.

While BEAMS survival is sensitive to higher early temperatures and thermal evolution, real-Earth mantle cooling history remains under debate. Petrological constraints indicate that Archean mantle temperatures may have been $\sim 100-300$ K higher than today (Herzberg et al., 2010). Our cases with HPEs predict significantly higher mantle temperatures compared to available constraints, both in the Archean and in the present day (see Fig. C5). Indeed, even in our cases without HPEs, for most of the period between 3.5 Ga to present, the mantle is hotter than what is realistic for Earth: from this perspective, the predictions of

our models (even those without internal heating) may be considered conservative. Future studies aimed at exploring the effects of radiogenic heating on BEAMS should address these discrepancies, for example by considering lower initial or boundary thermal conditions.

Furthermore, higher early mantle temperatures would enable phase transitions with different Clapeyron slopes than in the present day, enhancing or impeding mass fluxes between lower and upper mantle (Faccenda and Dal Zilio, 2017; Li et al., 2025) – potentially affecting BEAMS erosion. Nevertheless, BMO crystallization may have lasted billions of years and, if BEAMS are conceived as the product of BMO crystallization (Ballmer et al., 2025), then BEAMS erosion in an already cooled mantle would be diminished, also because less time would be available for entrainment and mixing until present.

While our models use a 2D spherical annulus geometry and exclude toroidal components of 3D mantle flow, implications for BEAMS preservation may be nuanced. On one hand, coherent blobs may be difficult to preserve in fully 3D flow fields that include both toroidal and poloidal components (Ferrachat and Ricard, 1998); on the other hand, viscous blobs may be more readily bypassed by mantle flow in 3D, enhancing their preservation (Merveilleux du Vignaux and Fleitout, 2001). Importantly, differences between 2D and 3D boundary layer instabilities do not strongly affect mixing timescales or regimes in high Rayleigh number convection (e.g., Coltice and Schmalzl, 2006; O'Neill and Zhang, 2018). Indeed, preliminary 3D models with similar compositional layering and rheology as explored here and in Gülcher et al. (2021) suggest that mixing is more inefficient and delayed, primarily due to dynamic differences in up/downwellings (Gülcher, 2022). In 3D geometry, variations in pile geometry due to intrinsic viscosity (McNamara and Zhong, 2004) may also influence BEAMS preservation and morphology, which remains subject to future research.

Segregation and entrainment of ROC and BEAMS are sensitive to model resolution (Tackley, 2011). Lower model resolution generally leads to increased mixing, due to under/overestimated segregation/entrainment, respectively (Tackley, 2011; Van Der Wiel et al., 2024; Heyn et al., 2018): indeed, Gülcher et al. (2021) show that higher resolutions promote BEAMS preservation, as entrainment is more accurately modeled. Similarly, higher resolutions enhance ROC accumulation (Tackley, 2011; Yan et al., 2020), indirectly supporting BEAMS survival. From this perspective, our model predictions in terms of BEAMS preservation remain conservative, and, along with 3D effects, higher resolutions may partly compensate the HPE effects described above.

## 5   Conclusions

In this study, numerical experiments are performed to investigate how the physical properties of recycled oceanic crust affect the long-term preservation of primordial material. Our main results can be summarized as follows:

- Primordial material preservation is enhanced by both intrinsic density and intrinsic viscosity of subducted crustal material (ROC).

- Survival of large, mid-mantle primordial blobs (BEAMS) is obtained only if recycled oceanic crust is sufficiently dense to form thermochemical piles (or a global layer) at the core-mantle boundary (CMB).

- For intermediate values of pile CMB coverage, high crust viscosity is critical to stabilize primordial blobs over geologic timescales.

As shown by our models, BEAMS preservation is profoundly affected by subducted recycled crust and its physical properties, highlighting the complex interaction between ROC and primordial, Bm-rich materials. In particular, the preservation of ancient, large-scale domains in the mid-mantle is not only compatible with, but also favored by deep (and viscous) thermochemical piles: seismic observations that confirm the presence of compositional heterogeneity above the core-mantle boundary are thus consistent with BEAMS survival in the present-day Earth mantle (Talavera-Soza et al., 2025). Coexistence of primordial and recycled heterogeneity is in agreement with geophysical and geochemical evidence, reconciling the paradigm of whole-mantle convection with the survival of chemically isolated geochemical reservoirs with potentially distinct isotopic signatures. Finally, the survival of ancient domains in the lower mantle may shift the bulk silicate Earth composition towards solar-chondritic values, reconciling cosmochemical evidence with petrological data for a pyrolitic upper-mantle composition.

*Code and data availability.* The numerical experiments presented in this paper are performed using StagYY (Tackley, 2008). Model output is accessible from an online repository (Desiderio, 2025). The StagPy suite (Morison et al., 2024) is used for data analysis. All figures are created using Matplotlib (Hunter, 2007), using scientific colormaps (Crameri, 2023).

*Video supplement.* Video supplements are accessible from an online repository (Desiderio, 2025).

## Appendix A: Additional Model Details and Parameters Explored

Here, we report general physical properties and parameters used in the models (see table A1).

Further, to ensure that the pyrolite density profile does not vary as $\Delta\rho_{Bs}/\rho_{Py}$ is changed between cases, the density jump at the 720 km phase transition for olivine $\Delta\rho_{Ol}$ is determined as follows:

$$\Delta\rho_{Ol} = (\Delta\rho_{Py} - F_{Px-Gt}\Delta\rho_{Px-Gt})/(1 - F_{Px-Gt}) \tag{A1}$$

where $F_{Px-Gt}$ is the fraction of *Px-Gt* in pyrolite and the constant $\Delta\rho_{Py}$ is the reference density jump for the pyrolitic mixture at the upper-lower mantle boundary from Gülcher et al. (2021).

To ensure that the pyrolite viscosity profile does not vary as $\zeta$ is changed between cases, the viscosity jump at the 720 km phase transition for olivine $\lambda_{Ol}$ is determined as follows, for any given $\zeta$:

$$\lambda_{Px-Gt} = \zeta\lambda_{LM} \tag{A2}$$
$$\lambda_{LM} = (\lambda_{Ol})^{F_{Ol}}(\lambda_{Px-Gt})^{F_{Px-Gt}} \tag{A3}$$

where $\lambda_{LM}$ is the lower-mantle background viscosity for the pyrolitic mixture (see table A1).

Additionally, we show depth-profiles for the density contrasts between pyrolite and other mantle materials in Fig. A1, calculated along the respective adiabatic temperature profiles. The colored areas indicate the ranges explored in the study, as explained in Sect. 2. Viscosity depth-profiles for all relevant mantle materials, calculated along a reference adiabat, are also shown in Fig. A2.

**Table A1.** Physical properties used in this study, based on the models of Gülcher et al. (2021). UM stands for upper mantle, LM stands for lower mantle, and PPV stands for post-perovskite. The adiabatic temperature, thermal conductivity, thermal expansivity, density vary with pressure, obeying a third-order Birch–Murnaghan equation of state Tackley et al. (2013); Gülcher et al. (2021).

| Property | Symbol | Value | Units |
|---|---|---|---|
| Mantle domain thickness | $D$ | 2890 | km |
| Gravitational acceleration | $g$ | 9.81 | m s$^{-2}$ |
| Surface temperature | $T_{top}$ | 300 | K |
| CMB temperature | $T_{CMB}$ | 4000 | K |
| Reference viscosity | $\eta_0$ | $5 \times 10^{20}$ | Pa s |
| Upper-mantle background viscosity | $\lambda_{UM}$ | 1 | |
| Lower-mantle background viscosity | $\lambda_{LM}$ | 1 | |
| PPV background viscosity | $\lambda_{PPV}$ | $10^{-3}$ | |
| Prim lower-mantle viscosity contrast | $\lambda_{Prim}$ | $10^2$ | |
| Reference temperature | $T_0$ | 1600 | K |
| Initial reference temperature | $T_{0,ini}$ | 1900 | K |
| Activation energy – UM and LM | $E_a$ | 140 | kJ mol$^{-1}$ |
| Activation energy – PPV | $E_{a,PPV}$ | 100 | kJ mol$^{-1}$ |
| Activation volume – UM and LM | $V_a$ | $1.8 \times 10^{-6}$ | cm$^3$mol$^{-1}$ |
| Activation volume – PPV | $V_{a,PPV}$ | $1.4 \times 10^{-6}$ | cm$^3$mol$^{-1}$ |
| Surface yield stress | $\sigma_0$ | 30 | MPa |
| Yield stress pressure derivative | $\sigma'$ | 0.01 | |
| Specific heat capacity | $C_P$ | 1200 | J (kg K)$^{-1}$ |
| Surface thermal conductivity | $k_0$ | 3 | W (m K)$^{-1}$ |
| Surface thermal expansivity | $\alpha_0$ | $3 \times 10^{-5}$ | K$^{-1}$ |

## Appendix B: Additional Figures

Here we include the color-map used for representing the three main compositional end-members (Basalt, Harzburgite and Primordial) and any intermediate mixtures between them.

Further, we show 2D histograms of Bs fraction $f_{Bs}$ and vertical velocity $u_z$, demonstrating the effect of $\Delta\rho_{Bs}/\rho_{Py}$ and $\zeta$ on both Bs enrichment in the deep mantle and vigour of convection within Bs-rich domains, and complementing Fig. 7. Only the lowermost $\approx 400$ km of the mantle are considered, so as to capture only the dynamics of the piles (where extant). These histograms simultaneously convey the degree of basalt enrichment in the lower mantle and how vigorous the convection is within enriched and non-enriched regions. As expected, the distributions for all cases in sub-regime M0 are similarly distributed around low $f_{Bs}$ and reach high $u_z$, i.e., the lowermost mantle is pyrolitic and is an integral part of whole-mantle convection. As expected, $\zeta$ has a minimal effect. Sub-regime MP is instead characterized by a "primary" peak at higher $f_{Bs}$, in addition

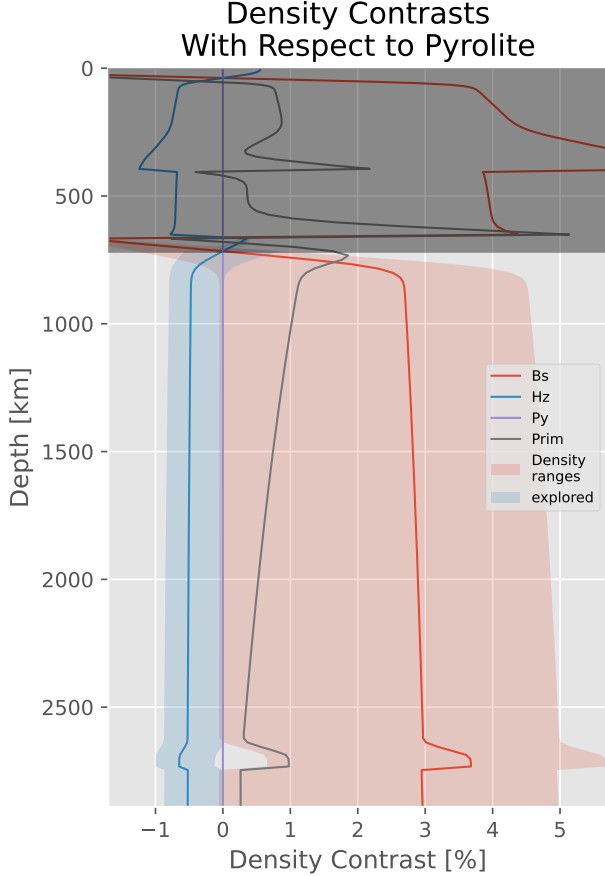

**Figure A1.** Profiles of density contrasts computed along the reference adiabat for Hz, Bs and Prim with respect to pyrolite. Solid lines denote the profiles used in the reference case. Shaded areas indicate the range explored in this study. The mantle below 720 km is highlighted in this plot for clarity.

to a "secondary" peak at lower $f_{Bs}$. This bimodal distribution can be interpreted as piles and "clear" areas above the CMB. When increasing $\zeta$, the primary peak is progressively narrowed along the $u_z$ axis, suggesting a reduction of convective vigour within the piles and a transition towards complete stratification, leading to regime B. Finally, as we increase $\Delta\rho_{Bs}/\rho_{Py}$ and

transition from sub-regimes MP, BP to sub-regime BL, the secondary peak becomes negligible and vigour of convection is further decreased (note the narrow distribution centered at $u_z = 0$ cm/y and the change of scale for the $u_z$ axis). This signifies the formation of a chemically stratified bottom layer. As expected, in sub-regime BL, the primary peak is increasingly narrowed down as a result of increasing $\zeta$. This is consistent with decreasing local Rayleigh number within largely stratified ROC heterogeneities.

Figs. B3 and B4 show temperature profiles taken at different azimuths (and at a single snapshot) to avoid averaging effects (similar to Desiderio and Ballmer, 2024). The two figures respectively show two pairs of cases (at t=4.5 Gy), one pair with

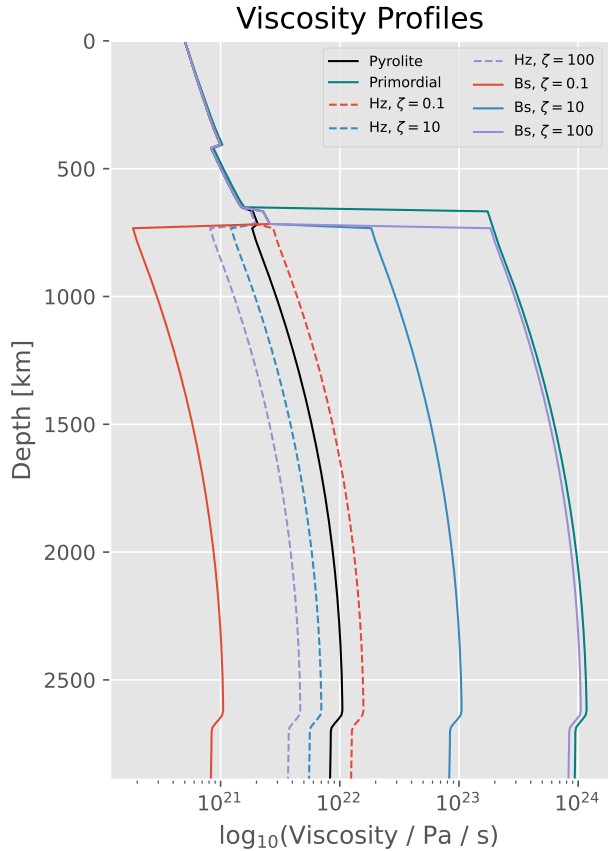

**Figure A2.** Viscosity profiles calculated along a reference adiabat for Hz, Bs, Py and Prim. The cases with $\zeta = 1$ are not plotted for visual clarity, as they would plot directly above the pyrolite profile.

ROC piles and the other with a ROC layer. For each pair, one model has low ROC viscosity and the other high viscosity. The larger annulus in the left-hand panels shows the temperature field, the smaller one shows $f_{Bs}$ (for $z > 2200$ km). Temperature profiles (in the right-hand panels) at different azimuths (shown with arrows of corresponding colors in the left-hand panels) are roughly adiabatic for the low ROC viscosity models, with TBLs above the CMB, and between the pile/layer and the mantle above. One single exception is the smaller pile for model ($\Delta\rho_{Bs}/\rho_{Py} = 2\%$, $\zeta = 0.1$), which is likely stratified. Temperature profiles for high ROC viscosity show a single thick TBL instead. Overall, these results support the conclusions of Sect. 3 in terms of convection/stagnation within the piles/layers as a function of ROC viscosity.


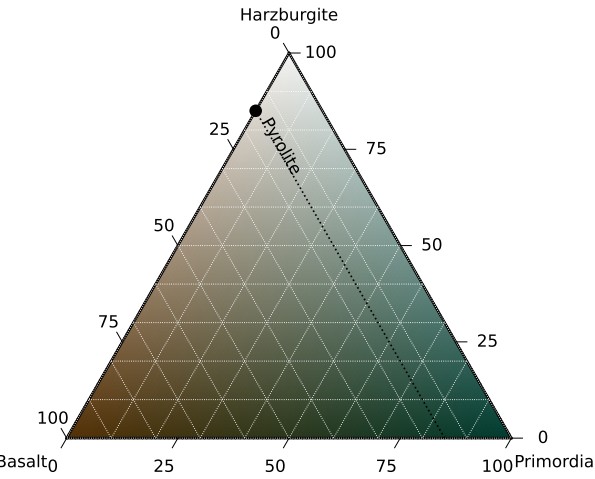

**Figure B1.** Ternary color-map used for representing the three main compositional end-members (Basalt, Harzburgite and Primordial)and intermediate mixtures. Pyrolite is also labeled for reference.

## Appendix C: Effects of internal heating

To explore the effects of internal heating (due to HPEs) on primordial-material preservation, we run four additional models where internal heating is generated in non-primordial tracers (which includes the pyrolitic noise distributed over the initial primordial layer: see Sect. 2). These additional cases (with 2%, 0.1; 2%, 1; 3%, 0.1; and 3%, 100 in terms of $\frac{\Delta\rho_{Bs}}{\rho_{Py}}$ and $\zeta$, respectively) correspond to four cases without HPEs: of these four cases with no HPEs, two cases are within the MP regime (with low and neutral Bs viscosity), and two within the BL regime (with low and high Bs viscosity).

In all additional cases, early evolution is slightly affected by internal heating, compared to the original cases in the main suite (see temperatures in video supplement): the upper pyrolitic layer heats up over time due to radiogenic heating, while the lower primordial layer is primarily heated from below. Eventually, downwellings accumulating atop the primordial layer promote a global-scale compositional overturn, cutting through the lower layer and forming initial BEAMS, similar to the cases described above (see Sect. 3.1).

Fig. C1 shows 4.5 Gy snapshots of the four cases with HPEs, while Fig. C2 compares the evolution of mantle temperature (averaged over the depth range of 700-900 km) between cases with and without HPEs: odd-numbered panels show temperatures within the ambient pyrolitic mantle (i.e. for $f_{Prim} < 0.5$, $f_{Bs} \leq 0.2$) and within BEAMS ($f_{Prim} \geq 0.5$). For all cases, HPEs increase ambient-mantle temperatures, thus also increasing convective vigour, compared to the analogous case without HPEs. Secondly, BEAMS lose heat less efficiently when the ambient mantle is internally heated: thus, BEAMS are generally less

viscous with HPEs, compared to the same case with HPEs. Third, downwellings are more vigorous in an internally heated mantle (compared to a case heated from the bottom), leading to more efficient fragmentation of BEAMS (Limare et al., 2019). This is reflected in Fig. C3, showing $f_{Prim}$ as function of time: here, cases with HPEs display an earlier overturn than the

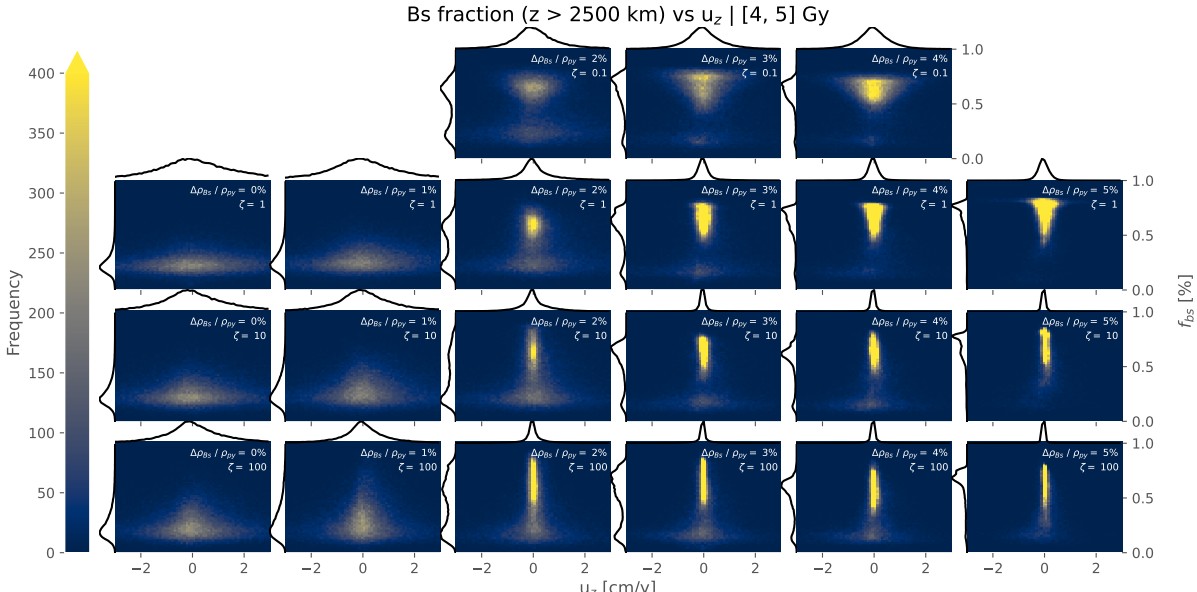

**Figure B2.** 2D histograms of Bs fraction $f_{Bs}$ and vertical velocity $u_z$ in the lowermost $\approx 400$ km of the mantle (represented on the horizontal and vertical axes respectively). Each panel refers to a different combination of parameters. Colors indicate the number of occurrences (counted in $t = 4.5 \pm 0.5$ Gy) and the color scale is saturated for clarity. Marginal distributions are qualitatively represented outside the main plots and are normalized to the respective maximum peak values.

corresponding case without HPEs, albeit a less vigorous one due to less extensive accumulation of downwellings. However, eventually, BEAMS erosion rates become faster for the cases with HPEs. BEAMS for the cases in with HPEs in Fig. C1 are
generally smaller than the equivalent cases without HPEs (see Fig. 2).

However, even in cases with internal heating, higher Bs densities tend to enhance BEAMS preservation (see Fig. C1). Also, Fig. C4 compares $f_{Prim}$ depth-profiles (averaged in the last Gy of model evolution) between cases with and without HPEs: while the average $f_{Prim}$ is roughly halved when internal heating is included in the model, the two internally-heated cases with $\frac{\Delta \rho_{Bs}}{\rho_{Py}} = 3\%$ have a slightly higher $f_{Prim}$ than all cases in regimes M0 and MP from the main suite (compare with Fig. 4).
Fig. C5 compares average mantle temperature evolution (in the depth range 560-760 km) between cases with and without HPEs (with no distinction based on composition as previous Fig. C2) and petrological constraints. The shaded gray area depicts the range of possible cooling trends since the Archean (Herzberg et al., 2010), starting from a present-day temperature at 660 km of 1873 K (Waszek et al., 2021). In all cases, model temperatures are higher than the petrological constraints from Herzberg et al. (2010), especially for the cases with HPEs.
Internal heating also enhances Bs accumulation at the CMB, since higher mantle temperatures facilitate Bs segregation from the subducted slab (Yan et al., 2020), such that in all four cases a Bs layer is stabilized at the CMB (see video supplement and Fig. C1). The effects of Bs intrinsic viscosity from the main study are preserved when internal heating is included: low Bs

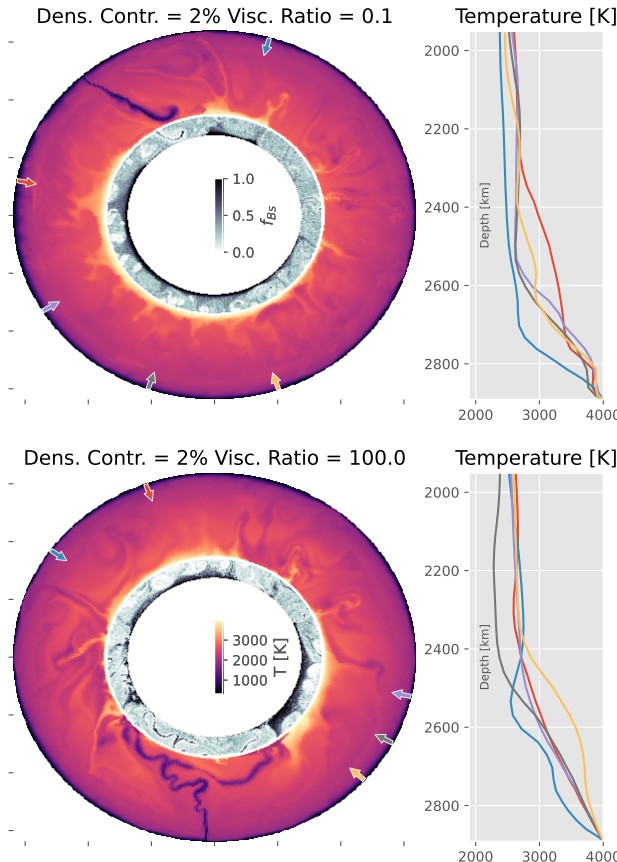

**Figure B3.** Temperature $T$ and Bs fraction $f_{Bs}$ snapshots at $t = 4.5$ Gy (larger and smaller annulus, respectively), along with temperature depth-profiles at different azimuths, for cases with convecting and stratified Bs piles (top and bottom row, respectively). The depth-range for the smaller annulus is $z > 2200$ km. The colored arrows denote the azimuths from which the profiles with the respective color are taken.

intrinsic viscosity promotes internal convection in the layer (see velocity fields in video supplement), although HPE-enrichment may enhance internal mixing rates (Citron et al., 2020).

Finally, Fig. C6 shows CMB heat-flux evolutions for all our cases, with and without HPEs. The figure supports the trends with intrinsic density and viscosity described in Sect. 3 of the main text (i.e., both physical properties tend to reduce CMB heat-flux). Cases with HPEs have a lower heat-flux than the equivalent case without HPEs, because of more efficient Bs accumulation as explained above.

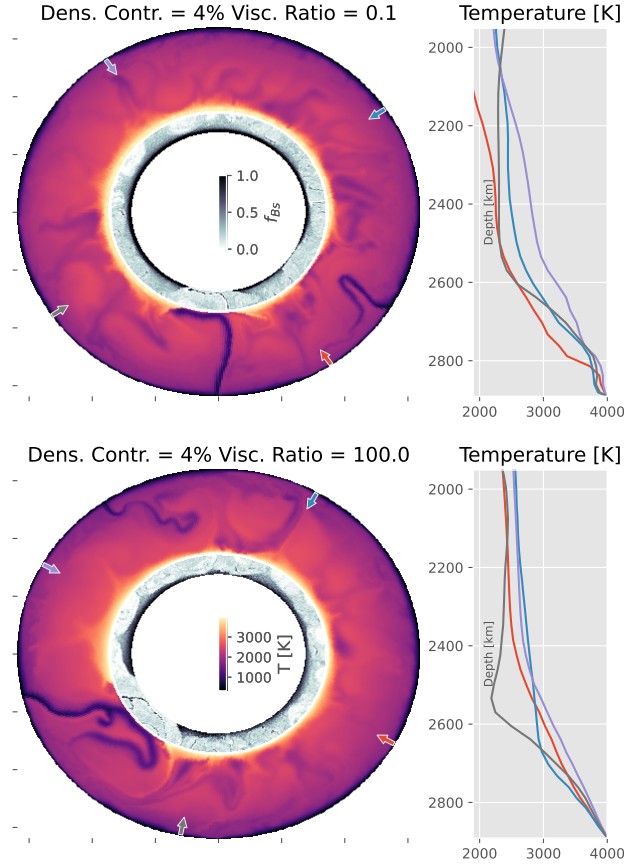

**Figure B4.** Same as in Fig. B3, but for two cases with a convecting and a stratified Bs layer (top and bottom row, respectively).

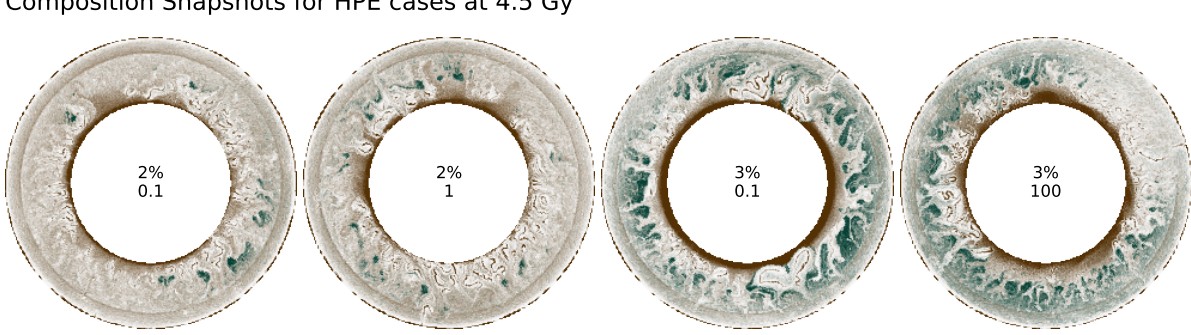

**Figure C1.** Snapshots of composition taken at $t = 4.5$ Gy of model evolution for cases with internal heating. The colormap is reproduced in Fig. B1. Controlling parameters of each case, $\Delta\rho_{Bs}/\rho_{Py}$ and $\zeta$, respectively, are labelled in the figure.

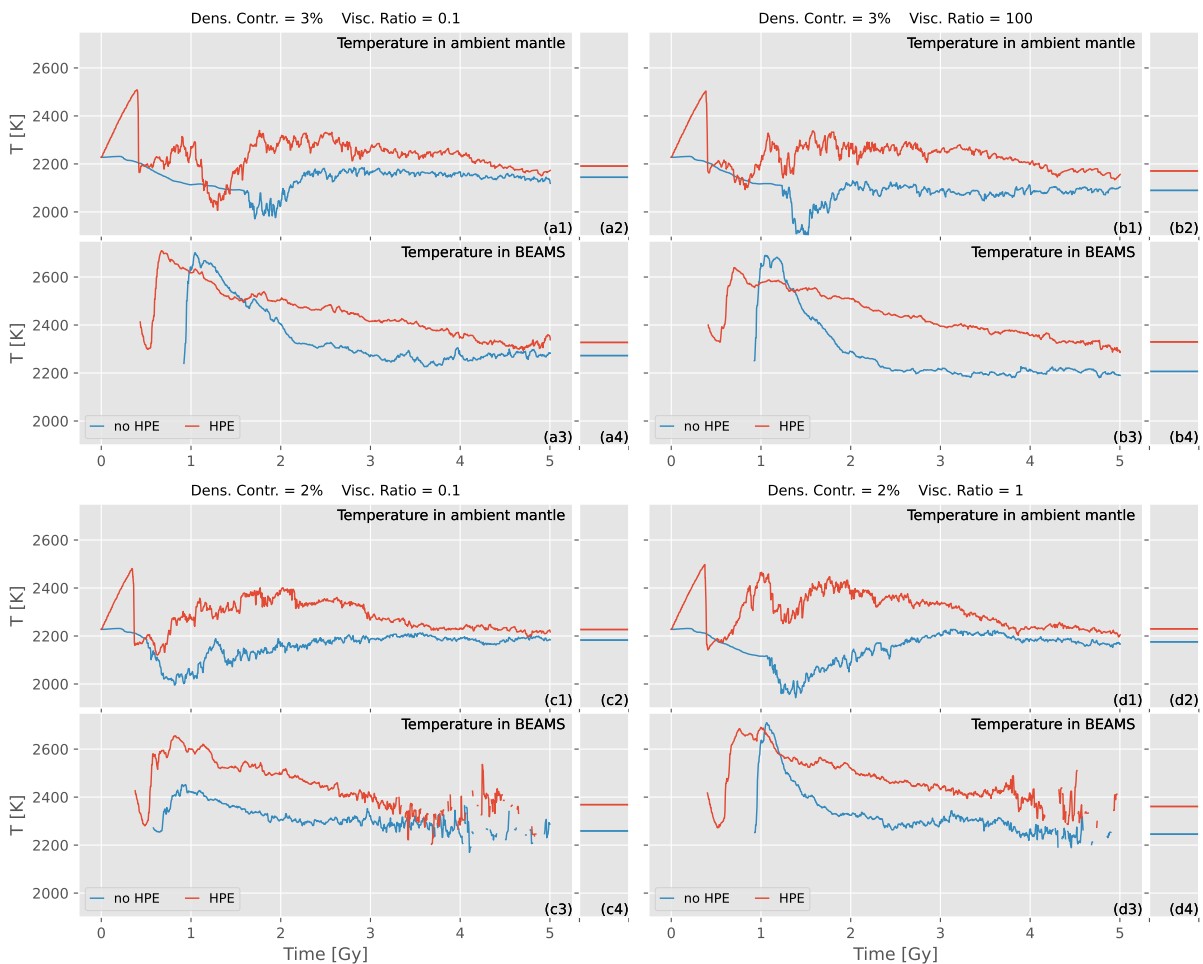

**Figure C2.** Mean temperature over time for cases with (red line) and without HPEs (blue line), in the ambient mantle and within primordial blobs (sub-panels 1 and 3, respectively). Primordial blobs are defined using the threshold $f_{Prim} > 0.5$. Ambient mantle is defined using the threshold $f_{Prim} \leq 0.5$ and $f_{Bs} \leq 0.2$. Gaps in the temperature in sub-panels c3, d3 are due to a lack of primordial material in the prescribed depth-range (700-900 km). Average temperature in the last Gy of model evolution is also shown in even-numbered sub-panels. Controlling parameters of the four cases shown as labelled.

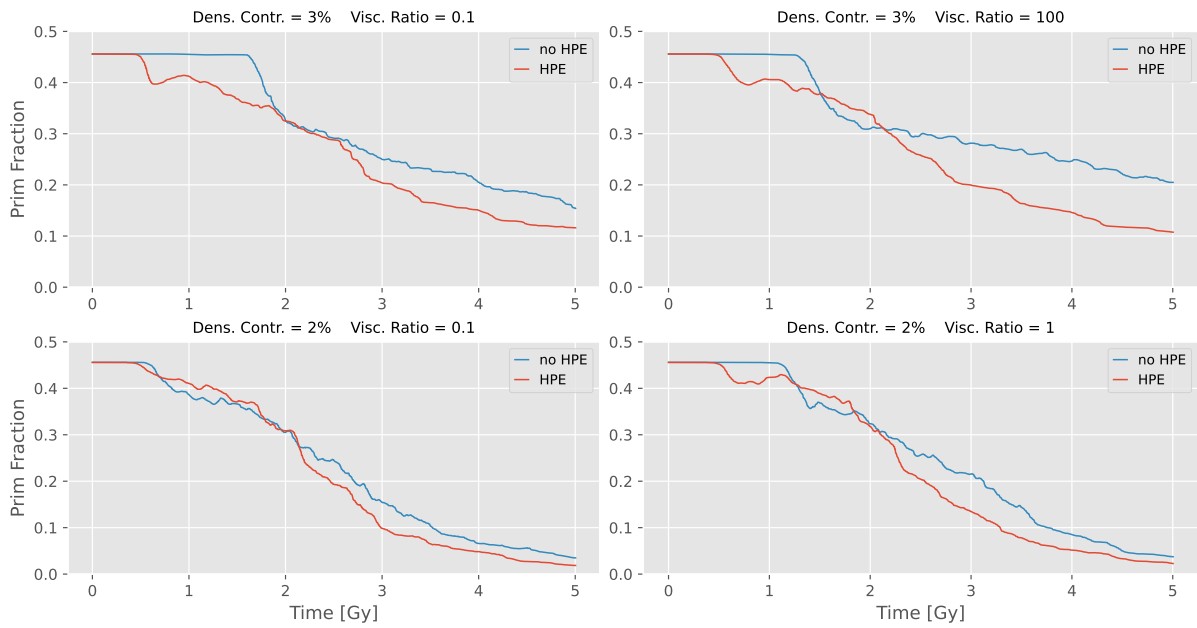

**Figure C3.** Mean primordial fraction $f_{Prim}$ over time, for cases with (red line) and without HPEs (blue line). Controlling parameters of the four cases shown as labelled.

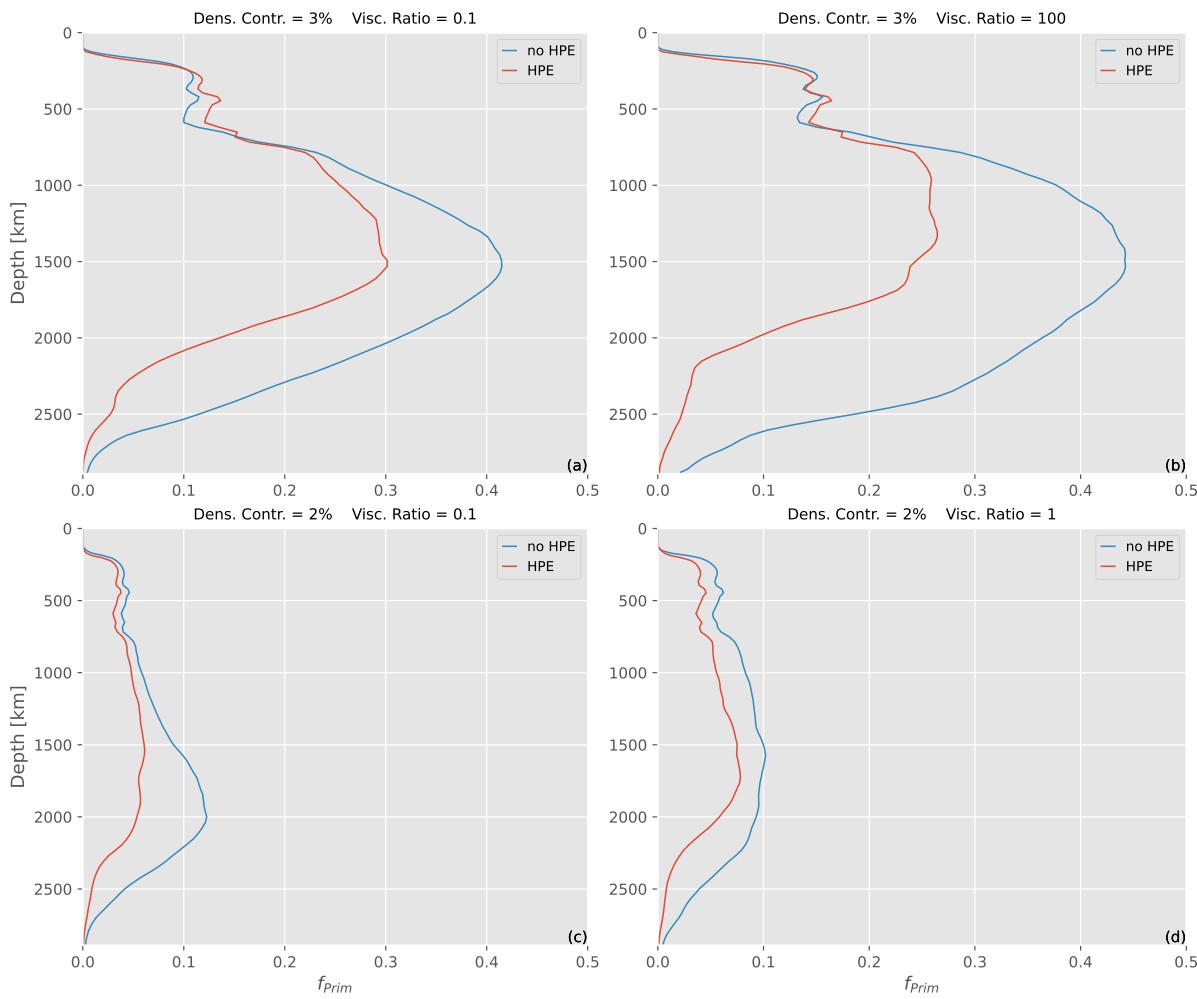

**Figure C4.** Depth-profiles of primordial fraction $f_{Prim}$, averaged over the last 1.0 Gy of model evolution, for cases with (red line) and without HPEs (blue line). Controlling parameters of the four cases shown as labelled.

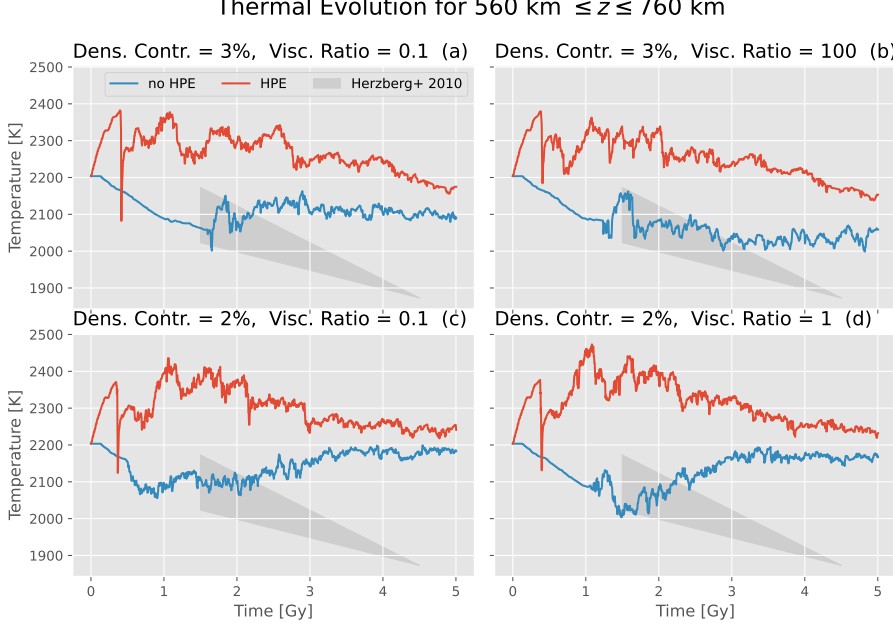

**Figure C5.** Mean mantle temperature over time, averaged over the 560-760 km depth interval, for cases with (red line) and without HPEs (blue line). Controlling parameters of the four cases are shown as labelled. The gray shaded area represents the range of possible temperature evolutions from the Archean to the present-day based on petrological estimates (Herzberg et al., 2010). The petrological estimates for mantle temperatures at the base of the lithosphere are projected to the relevant depth by considering a present-day temperature of 1873 K at the 660 km discontinuity (Waszek et al., 2021).

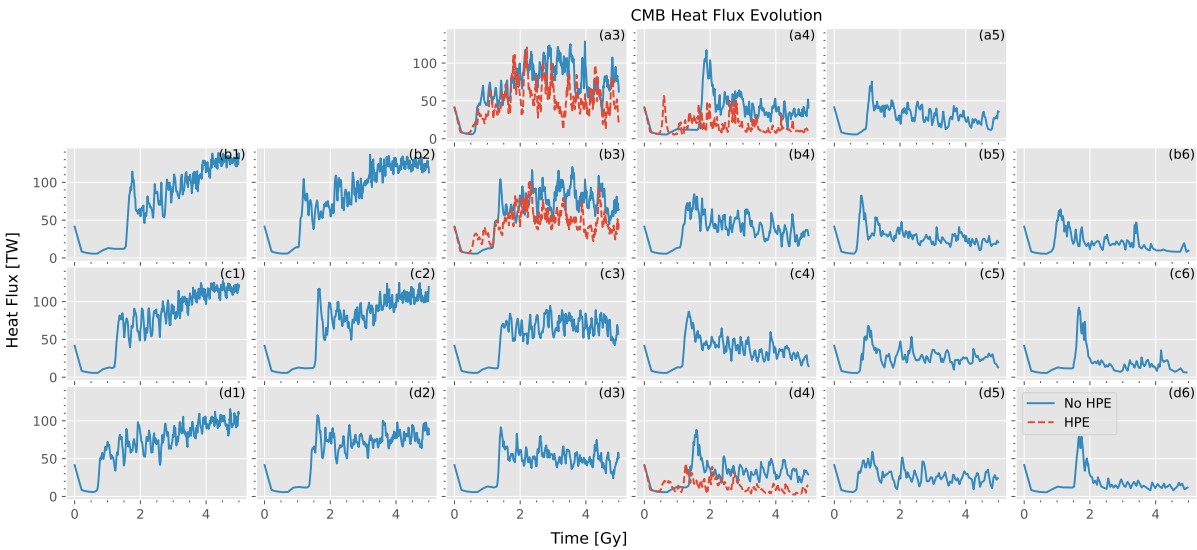

**Figure C6.** Core-mantle boundary heat-flux over time, for cases with (red line) and without HPEs (blue line). Panel labels a-d correspond to viscosity ratio values $\zeta = 0.1 - 100$, panel labels 1-6 correspond to density excess values $\frac{\Delta \rho_{Bs}}{\rho_{Py}} = 0\% - 5\%$.

*Author contributions.* Conceptualization, Methodology, Formal Analysis, Visualization, Writing (Original Draft Preparation; Review and Editing): M. D. Conceptualization, Methodology, Writing (Review and Editing): A. J. P. G. Supervision, Resources, Conceptualization, and Writing (Review and Editing): M. D. B.

*Competing interests.* The authors declare that they have no conflict of interest.

*Acknowledgements.* M.D.B. acknowledges NERC standard grant NE/X000508/1. M.D. acknowledges support from the Department of Earth Sciences at University College London and from the UPFLOW project, funded by the European Research Council under the European Union's Horizon 2020 research and innovation program (grant 101001601). We thank the associate editor Juliane Dannberg and both reviewers, whose comments helped improve this work. M.D. would further like to thank Ana Ferreira, Andrew Thomson and John Brodholt for their valuable feedback. Computations have been run on the Euler cluster at ETH Zürich. A.G. acknowledges funding from the Center for Space and Habitability (CSH) at the University of Bern as well as NCCR PlanetS supported by the Swiss National Science Foundation under grant 51NF40_205606.

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
