# Peer review of "Primordial-material preservation and Earth lower-mantle structure: the influence of recycled oceanic crust"

_EGUsphere, 2025_

## Referee Comment (RC2)

In this article, the authors investigate the influence of recycled oceanic crust properties (density excess and viscosity) on the preservation of lower Earth mantle primordial heterogeneities, consisting of bridgmanite material resulting from the crystallization of the Earth early magma ocean, and which are referred to as BEAMS. For this, they perform more than 20 simulations of thermo-chemical convection with three sorts of material, including basalts, harzburgite and primordial material (BEAMS), and varying both the viscosity and the density of basalts. After 4.5 Gyr, simulations may be sorted out in 4 regimes or sub-regimes depending on the evolution of the BEAMS and recycled basalts. These include full mixing of both basalts and BEAMS with harzburgite, mixing of BEAMS with the formations of basal piles of basalts, and preservation of BEAMS either with the formation of piles of basalts or with basaltic layering above the CMB. The authors note that the occurrence of each regime is controlled by the excess density of basalts and, to a lesser extent, by their viscosity. They conclude that dense recycled basalts is needed to preserve BEAMS over periods of time comparable to the age of the Earth, and that higher viscosity further helps this preservation.

This article is well written, and the research it present fits well the scope of *Solid Earth*. The simulations of convection are clearly described and are carefully performed with a state-of-the code. Results and interpretation are also clearly discussed and are supported by the authors simulations. I have only minor comments, mainly points of discussion on BEAMS, and I recommend this article for a publication in *Solid Earth* after some minor to moderate revisions. Below are some comments and suggestions that the authors may include in the revised version of their paper.

1. The authors neglect internal heating. This is fine, except that they probably underestimate the potential effect of internal heating, and that this aspect should deserve more discussion. Adding internal heating is likely to affect the balance between plumes rising from the CMB and descending slabs in the favor of the later. I guess that more vigorous slabs may have some impact on the preservation of BEAMS. Second, it is (as the authors pointed out) likely that heat producing elements will concentrate in basalts. This may in turn affect the evolution of basaltic piles or layering in a way similar to the impact of excess heating in primordial material (as recently investigated by Guerrero et al., 2024). Finally, adding internal heating is important for the mantle heat budget and in particular to get CMB heat flux within the expected range of values. Again, no additional simulation is needed here, but an extended discussion on that topic would be welcome.

2. To separate the different regimes, the authors use threshold values of the local fractions in primordial material and basalts, and in the fraction of CMB area covered by basalts. Are they any specific reasons for fixing these parameters to the values chosen by the authors (if yes, explain), or are these values mostly arbitrary ?

3. I find the discussion on the characteristic viscosity for each regime (page 10) and the corresponding figure 4 not very convincing. More precisely, when accounting for the variability of viscosity within each regime, the difference between two different regimes sounds less

significant. I agree that there is a general trend (models that preserve BEAMS have on average higher viscosity), but the most viscous model in MP have nearly the same viscosity than the less viscous model in BL, so I'm not sure that these profiles are a key aspect of the authors results. Also, MP and MO regimes have less simulations than BL, and I guess than running more models would enlarge the range of viscosity. Finally, it would be interesting to show (or say) which model correspond to the upper and lower bounds of each radial viscosity profiles in figure 4.

4. Line 286. $q_{bot}$ (bottom heat flux, I guess), has not yet been defined in the paper. It might also be interesting to plot the evolution of $q_{bot}$ and $q_{top}$ with time.

5. The authors point out that the basal basaltic layer is animated by convection. I'm not sure this is the case. The fact that the radial velocity is not zero and vary laterally within these structures does not guarantee that convection operates in them. First the authors should compare these velocities with those in the mantle. Second, and more importantly, if the basal layer is animated by convection, the temperature profile within this layer should consist of a thermal boundary layer at both the top and the bottom and an adiabatic region in between. If the temperature profile is linear, then the layer is not animated by convection, and heat is transported by conduction. Same thing for the piles of basalts. For similar reasons, on line 345, I would change 'reduces convective vigor within the piles' to 'reduces heat transfer within the piles'.

6. I'm less optimistic than the authors regarding the seismic signatures of BEAMS. First, BEAMS are large structures, so I guess they would be resolved by the most recent tomographic maps, especially those based on waveform inversion. Second, while excess bridgmanite have no or very slight effect on shear-velocity, the BEAMS enrichment in iron (~ 2-3%) should produce a relatively strong signature (namely, a substantially reduced shear-wave velocity), enhanced by the fact that BEAMS are slightly hotter than average mantle. Overall, I do not see any reasons why BEAMS shouldn't be detected by available tomographic models.

---

## Author Comment (AC1)

**RESPONSE TO REVIEW**

**REVIEWER #1**

The study by Desiderio et al. presents a systematic exploration of the effect of viscosity and density differences on the preservation of primordial mantle material as BEAMS and the generation of LLSVP-like accumulations of recycled oceanic crust (ROC). Overall, I think that this is a nice contribution and should be suitable for publication after minor revisions. In my view the most significant issue is the omission of internal heating, which at least needs to be physically justified.

We thank the reviewer for their time and their insightful feedback. We address the comments one by one in the following.

1) Line 28: I do not think that it's reasonable any longer to claim that the nature of the LLSVPs is strongly debated. Masters et al. (2000) described the anti-correlation of bulk and shear wavespeeds in the lowermost mantle and there is no way to reconcile this with a purely thermal origin. More recent work by Lau et al. (2017) using tidal constraints and by Moulik and Ekstrom (2016) using normal mode and body wave tomography make a compelling case for high intrinsic density within (at least part of) the LLSVPs as well. It seems that this debate was vigorous in the 2000s but has been long since settled.

We rephrased the sentence (see new manuscript in lines 27-29).

2) Line 48: Seismic-tomography images is awkward. Suggest "tomographic images" instead.

We changed as suggested (line 49).

3) Line 120: The choice to run models without internal heating needs to be justified physically as it seems inconsistent with what we know about the energetics of mantle convection, and the paper aims to understand the preservation of heterogeneity under Earth-like mantle evolutionary scenarios. Under ancient, hotter mantle conditions, mantle heat production was larger than the present day value and the omission of mantle heat production seems even more unreasonable. In general, heat production will destabilize convectively isolated 'blobs' in the mantle because they will heat up and become more buoyant and less viscous (e.g. Becker et al., 1999). It's not clear to me, given all of the complexities of composition-dependent buoyancy and viscosity considered here, how the choice

to run models that are not earth-like in terms of the energetics of convection can be justified. I understand that HPEs are incompatible in Bridgmanite, so they may be concentrated in the later products of magma ocean crystallization. But they are also even more incompatible in ferropericlase and moderately incompatible in CaPv, and they have to go somewhere to conserve mass during magma ocean crystallization! The recycled oceanic crust should also be enriched in HPEs and the harzburgitic residue depleted. The authors could comment on whether this could affect the dynamics of ROC accumulation in the lowermost mantle.

To address both BEAMS survival and ROC accumulation in an internally heated mantle, we ran 4 additional cases and describe the results at the end of the Results section (newly added lines 332-355) and in a newly added Appendix C (starting at line 619-620, plus figures C1-5) - accompanied by an updated video supplement. We also expanded the Discussion accordingly (lines 516-531). We finally point out that, under the bottomupwards crystallization scenario (implied by our initial layered setup), a negligible fraction of fp and ca-pv cumulates are expected to crystallize in the lower mantle (e.g, see: Boukare' et al., 2015; Caracas et al., 2019; Naibiei et al. 2021). Even for 50% crystallization ('batch crystallization'), almost exclusively bm crystallizes (Caracas et al., 2019). HPEs would then remain in the melt that later crystallizes to become the upper mantle. In other words, for any MO crystallization scenario, the compatibility of HPE in fp and ca-pv does not matter (because they do not crystallize); it is the partitioning coefficient between bm and melt that matters. Later on, after the initial global overturn, the HPEs would be confined in the ambient mantle that circulates around the BEAMS (as predicted by our models), and BEAMS would not be internally heated (unlike the blobs modeled in Becker et al., 1999).

4) Line 130: The treatment of basalt/eclogite buoyancy in the deep mantle is simplified and the dynamics may be quite different if the authors used thermodynamic lookup tables for pyrolytic and basaltic compositions from Stixrude and Lithgow Bertelloni (2024)

We added lines 381-385 to the manuscript to discuss this point.

In addition, we would like to emphasize that we explored density contrasts over a wide range of values, which encompass experimental uncertainties, including the range of thermodynamic databases such as *Stixrude and Lithgow Bertelloni* (2011) as well as *Stixrude and Lithgow Bertelloni* (2024).

The figure below shows density differences with respect to pyrolite computed using the models of *Stixrude and Lithgow Bertelloni (2011-2024)* - as indicated by the legend -, computed along isentropic profiles with potential temperature 1600K, using Perple\_X (Connolly 2005). Note that the scale on the x-axis is dln\_rho in %, thereby roughly

marking the range of values explored in this study. While the density contrasts obtained in the mid-mantle (~2%) are well within the parameter space explored in our study, the density anomaly systematically decreases towards the CMB for *Stixrude and Lithgow Bertelloni (2024)*: this is a different behaviour than what is used in our study, and may be due to the incorporation of more recent datapoints for compressibility in the database. This difference may change our results in terms of the style and location of Bs segregation. In terms of Bs stability, our case with dln\_rho=1% might be most consistent with *Stixrude and Lithgow Bertelloni (2024)*. Even if in this case no large thermochemical piles may be formed from recycled crust, we emphasize that our results with thermochemical piles would still be largely applicable, as long as the piles are formed by another mechanism not included in our models (e.g., BMO cumulates) - see new lines 381-385.

However, we also point out that thermodynamic databases such as the ones by Stixrude and Lithgow-Bertelloni are compilations of experimental and theoretical results that come with their own set of assumptions (and uncertainties, on the order of 2% for densities in the lower mantle, see e.g. Connolly and Khan 2016): indeed, using their model from 2011, we obtain profiles with shallower gradients with depth (see figure). Also, we highlight that the previous geodynamic studies that explored BEAMS (e.g. Gulcher et al 2021) preservation used density profiles designed to match the PREM model (Dziewonski and Anderson, 1981), similar to Tackley et al., 2013: we chose to keep these assumptions in our reference model for consistency with those studies.

5) Section 2.4: The temperature dependence of effective Clapeyron slopes is not considered. I wonder if the authors might at least comment on this? The equations of state developed by Stixrude and Lithgow-Bertelloni (2011, 2024) predict that the effective Clapeyron slope will change over time in a cooling mantle, which will affect the tendency towards layering and certainly the viability of BEAMS preservation.

We addressed this in a newly added paragraph in the Discussion (lines 533-537).

6) Line 170-175: These dynamics might be quite different if mantle heat production was included.

See newly added paragraph at the end of the Results section, newly added Appendix C (specifically lines 342-343, 626-630) and video supplement. Also see response to comment #3

7) Line 197: The mathematical symbol used here is showing up as both a greater than and less than sign, which I find confusing. Why not just define a threshold of f=0.3? Presumably the results are not very sensitive to the choice of this value?

We agree that the symbol is source of confusion (we use indeed a threshold of 0.3), we thank the reviewer for pointing that out. We changed lines 197-199 (see new lines 199-201). We also changed line 211 to: "sub-regimes MP and BP correspond to  $\zeta \le 10$  and  $\zeta \ge 10$  respectively" - see new line 213.

8) Figure 3: I found this figure extremely confusing. There's too much going on for me to understand quickly what's being shown and it would almost be easier to look at a data table – the opposite of what a figure is supposed to accomplish. I would find it clearer to have four separate panels with contour plots, and perhaps use the same color scale for all four panels to highlight trends. It's too hard for me to see the trends in the four different quantities on the same plot with four different symbols and four different color maps. Also, why abbreviate everything? The figure would be much more interpretable if the authors show the regime boundaries and give each region a nice simple label like 'Mixed Primordial' or 'deep ROC layer'. You could also annotate the axes – buoyancy ratio and viscosity ratio.

We thank the reviewer for their helpful comments of how to better convey our results. We recreated figure 3 as suggested.

9) 212: I think that radially-averaged depth profiles is redundant and maybe incorrect. I would call this the azimuthal average or horizontal average.

Agreed, we changed it to simply "depth profiles" (line 214).

10) 213: regime -> regimes

We fixed the typo (215).

11) Section 4.3: It would be nice for the authors to also address the preservation of primordial noble gas signatures in the OIB source. Are the authors arguing that the BEAMS are a likely source of primordial noble gas (He, Ne, Xe) isotopes? If so, is this at odds with the idea that BEAMS represent the early crystallization products since the noble gases are highly incompatible during solidification?

We expanded the discussion of this point (lines 455-465). Noble gases are often assumed to be incompatible, but whether this is indeed the case at lower-mantle coditions is still under debate. Indeed, there are few experiments in this regard carried at P/T conditions that are relevant for (B)MO crystallization in the lower mantle (Moreira 2013, Karato 2016) and available experimental results are discordant (e.g., Shcheka & Keppler 2012, Jackson 2021, who respectively report high and low Ar solubility in Bm). In general, we present BEAMS as a viable hypothesis for a primordial geochemical reservoir, albeit one that needs further testing via additional measurements of noblegas solubilities/partition coefficients at high P and T. In the manuscript, we also posit an alternative pathway for BEAMS to host noble-gas signatures- if indeed these noble gases are highly incompatible- by considering interstial melt/melt inclusions in the (B)MO cumulates (see Jackson 2021).

13) 443: There is also a new global attenuation tomography paper (published while the present manuscript was in review) that shows an interesting feature around 1000 km depth: Sun et al. (2025) A high attenuation layer around 1000 km depth. Earth and Planetary Science Letters 669, 119577.

This an intriguing and relevant publication. We thank the reviewer for highlighting it. We now discuss it in the Discussion section (line 407-408).

---

## Author Comment (AC2)

**REVIEWER #2**

In this article, the authors investigate the influence of recycled oceanic crust properties (density excess and viscosity) on the preservation of lower Earth mantle primordial heterogeneities, consisting of bridgmanite material resulting from the crystallization of the Earth early magma ocean, and which are referred to as BEAMS. For this, they perform more than 20 simulations of thermo-chemical convection with three sorts of material, including basalts, harzburgite and primordial material (BEAMS), and varying both the viscosity and the density of basalts. After 4.5 Gyr, simulations may be sorted out in 4 regimes or sub-regimes depending on the evolution of the BEAMS and recycled basalts. These include full mixing of both basalts and BEAMS with harzburgite, mixing of BEAMS with the formations of basal piles of basalts, and preservation of BEAMS either with the formation of piles of basalts or with basaltic layering above the CMB. The authors note that the occurrence of each regime is controlled by the excess density of basalts and, to a lesser extent, by their viscosity. They conclude that dense recycled basalts is needed to preserve BEAMS over periods of time comparable to the age of the Earth, and that higher viscosity further helps this preservation. This article is well written, and the research it present fits well the scope of Solid Earth. The simulations of convection are clearly described and are carefully performed with a state-of-the code. Results and interpretation are also clearly discussed and are supported by the authors simulations. I have only minor comments, mainly points of discussion on BEAMS, and I recommend this article for a publication in Solid Earth after some minor to moderate revisions. Below are some comments and suggestions that the authors may include in the revised version of their paper.

We thank the reviewer for their work and for their valuable comments, that we address below.

1. The authors neglect internal heating. This is fine, except that they probably underestimate the potential effect of internal heating, and that this aspect should deserve more discussion. Adding internal heating is likely to affect the balance between plumes rising from the CMB and descending slabs in the favor of the later. I guess that more vigorous slabs may have some impact on the preservation of BEAMS. Second, it is (as the authors pointed out) likely that heat producing elements will concentrate in basalts. This may in turn affect the evolution of basaltic piles or layering in a way similar to the impact of excess heating in primordial material (as recently investigated by Guerrero et al., 2024). Finally, adding internal heating is important for the mantle heat budget and in particular to

get CMB heat flux within the expected range of values. Again, no additional simulation is needed here, but an extended discussion on that topic would be welcome.

To address these points, we ran 4 additional cases with internal heating and with HPE partitioning as suggested, now described at the end of the Results section (lines 332-335) and in detail in Appendix C, starting at line 620 (accompanied by an updated video supplement). We also expanded the Discussion accordingly (lines 516-531).

2. To separate the different regimes, the authors use threshold values of the local fractions in primordial material and basalts, and in the fraction of CMB area covered by basalts. Are they any specific reasons for fixing these parameters to the values chosen by the authors (if yes, explain), or are these values mostly arbitrary?

We selected those values as follows: first, we inspected the models visually to draw out boundaries between regimes, then we checked if these corresponded to reasonable numerical thresholds: we chose these values as boundaries between regimes to obtain a quantitative measure. These thresholds could be slightly varied within reasonable ranges without changing the regime boundaries.

3. I find the discussion on the characteristic viscosity for each regime (page 10) and the corresponding figure 4 not very convincing. More precisely, when accounting for the variability of viscosity within each regime, the difference between two different regimes sounds less significant. I agree that there is a general trend (models that preserve BEAMS have on average higher viscosity), but the most viscous model in MP have nearly the same viscosity than the less viscous model in BL, so I'm not sure that these profiles are a key aspect of the authors results. Also, MP and MO regimes have less simulations than BL, and I guess than running more models would enlarge the range of viscosity. Finally, it would be interesting to show (or say) which model correspond to the upper and lower bounds of each radial viscosity profiles in figure 4.

The viscosity profiles in figure 4 are reported for the sake of completeness and, indeed, are not used to support our discussion or drive any particular point about trends. Incidentally, we note that "the most viscous model in MP" and "the less viscous model in BL" are separated by at least a factor 2 (e.g. at 1500 km depth). The figure attached in this response also shows that the viscosities of M0 are similar to those of MP; the viscosity for BP is comparable to the lower end of the range shown for BL viscosities (the text has been updated to highlight this: see lines 292-294). We also update the caption of figure 4 with information about the upper and lower bounds of each visc. profile. The minimum and maximum boundary of the viscosity ranges in the figure correspond to models with:

dRho = 1%,  $\zeta$  = 1.0, and dRho = 1%,  $\zeta$  = 100.0;

dRho = 2%,  $\zeta$  = 0.1, and dRho = 2%,  $\zeta$  = 10.0; dRho = 3%,  $\zeta$  = 0.1, and dRho = 5%,  $\zeta$  = 1.0;

for regimes M0, MP and BL, respectively. Overall, this supports our point that ROC density and viscosity both enhance primordial preservation by reducing convective vigor.

4. Line 286. qbot (bottom heat flux, I guess), has not yet been defined in the paper. It might also be interesting to plot the evolution of qbot and q top with time.

We corrected this, see lines 288 and 289. We added this plot in appendix C (Fig C6).

5. The authors point out that the basal basaltic layer is animated by convection. I'm not sure this is the case. The fact that the radial velocity is not zero and vary laterally within these structures does not guarantee that convection operates in them. First the authors should compare these velocities with those in the mantle. Second, and more importantly, if the basal layer is animated by convection, the temperature profile within this layer should consist of a thermal boundary layer at both the top and the bottom and an adiabatic region in between. If the temperature profile is linear, then the layer is not animated by convection, and heat is transported by conduction. Same thing for the piles of basalts. For similar reasons,

**on line 345, I would change 'reduces convective vigor within the piles' to 'reduces heat transfer within the piles'.**

To check for convection, we visually inspected the movies (which are accessible via link in the video supplement), where convection in the Bs-enriched regions is apparent in the velocity field: in particular, convection cells in Bs-rich regions are visible in some models while absent in others (with our choice of colorscale), depending on viscosity (as described in the manuscript): see the vigorous vertical positive/negative velocities in the movies close to the CMB, inside the contours for composition. We note that the temperature profiles shown in the paper would not be suitable to observe the gradients described by the reviewer, given they are both spatial and temporal averages. To address the point brought up by the reviewer, we add Figs. B3 and B4 to appendix B (see lines 316-320, 611-618). The figures show the gradients described by the reviewer, supporting our conclusions in terms of convection in the piles/layers.

6. I'm less optimistic than the authors regarding the seismic signatures of BEAMS. First, BEAMS are large structures, so I guess they would be resolved by the most recent tomographic maps, especially those based on waveform inversion. Second, while excess bridgmanite have no or very slight effect on shear-velocity, the BEAMS enrichment in iron (~ 2-3%) should produce a relatively strong signature (namely, a substantially reduced shear-wave velocity), enhanced by the fact that BEAMS are slightly hotter than average mantle. Overall, I do not see any reasons why BEAMS shouldn't be detected by available tomographic models.

Our primordial material density profile is consistent with (Mg,Fe)SiO3 perovskite with a Fe# 0.12, equivalent to 6% FeO of molar fraction in Bridgmanite, similar to the molar fraction of FeO in pyrolite (see e.g. Table 1 from Xu et al. 2008): thus, BEAMS are not enriched in iron. Indeed, MgSiO3 perovskite is expected to be seismically 'faster' than the ambient mantle, at the same temperature (Wentzcovitch et al., 2004): this compositional effect on velocity may be compensated by the relatively warm temperature of the BEAMS (Gülcher et al. 2021).

More importantly, it tends to be difficult for tomography to resolve structures in the mid-mantle. We here attach some additional seismic-modelling results (in prep. for future publication) based on the predictions of our geodynamic models from this study. The attached snapshots represent seismic velocity anomalies associated to the pressure-temperature-compositional fields predicted by our numerical models, using the thermodynamic database by Stixrude & Lithgow Bertelloni, 2011 (via Gibbs energy minimization, Connolly 2008). The compositions for Hz & Bs are taken from Xu et al., 2008. These results show that a negative velocity anomaly is associated to BEAMS, but has a very low amplitude (~-1.5% at most). We assess whether these anomalies could be discriminated via mantle tomography by processing the models using 'filtering' operators (e.g. Ritsema et al., 2007) based on the SGLOBE-rani tomographic model

(Chang et al., 2015). Arrows indicate BEAMS. Case B, C correspond to model with visc ratio=10/dens.contr.=2% and visc ratio=1/dens.contr.=3%. Case A is a model from Desiderio and Ballmer, 2024 - with no prim. Odd/even rows depict the unfiltered/filtered snapshots. The first 2 rows test the effect of filtering in an optimal scenario where BEAMS occur in areas beneath dense seismic-station coverage (USA), as opposed to a low-coverage scenario (bottom 2 rows, Africa). As expected, anomalies are dampened and smeared (

---

## Author Response (AR2)

**Response to Editor Comments**

**Thank you for revising the manuscript and addressing the referee's comments so thoroughly. Before I accept the manuscript, I'd like to give you the opportunity to address some minor points I noticed when reading through the changes:**

We thank the Editor for her work and for her helpful comments, which we hope we can address satisfactorily. We respond point by point below.

**- Would it make sense to mention the HPEs as an additional parameter being explored early on, in the Methods section? (for example, in 2.4 Parameters Explored) As it is, it comes a bit as a surprise in 3.3.**

We changed the manuscript to introduce the cases with HPEs at the end of Sect. 2.4. We also moved the relevant methodological details from Sect. 3.3 to 2.4 (see lines 160-171 in the markup file). In Sect. 3.3, we now only focus on describing the results of this secondary suite (lines 341-353; 366-370, markup file).

**- It looks like some text in the caption of Figure 3 still refers to original version of the figure**

Thank you for spotting this, we corrected it

**- I would suggest to increase the font size of the labels for density change/viscosity contrast in figure 5 and 6, I found these quite hard to read.**

We increased the label size

**- line 322: acros --> across**

Corrected (line 326, markup file)

**- 340: Would it make sense to mention the regimes of these models here as well? (just briefly in parentheses to remind the reader)**

We added such a reminder (lines 345,346 in markup file)

**- 338 and following: I think it is great that you decided to not simply discuss the effect of HPEs but to run new models that demonstrate their effect. I think that gets the point across much better. There was one point I wanted to make though: It took reading the whole new section and the whole new Appendix C for me to realize that for the 4 new models, they had all changed their regime compared to their counterparts without HPEs, so that according to the criteria outlined above (fprim smaller than 0.3) they would now count as being in a completely new regime (ROC layer and no blobs). This of course changes how we should interpret the results, but on the other hand, you make the argument that the models with added HPEs**

**are way hotter than we expect Earth to have been in the past (and therefore the viscosity is lower compared to what we would expect). Since there are HPEs present in Earth's mantle, I think it would be good to clearly state these two points the end of section 3.3 (the change of regime and the limitation of the models being much hotter than we the Earth's mantle was), just to make it obvious to the reader, so that they know: Yes, HPEs do affect the regime boundaries, but their full effect is complicated and requires additional models beyond the scope of the current study. In addition, I think these results are also relevant to the discussion in Section 4.1 "Results summary and Relation to Previous Work", so I was surpised that they were not mentioned there.**

Thank you for pointing this out. We added lines 366-370 to clarify that 1) a different regime is obtained when HPE are considered, and 2) that the mantle is hotter than realistic, affecting the applicability of these four cases (we then refer the reader to the more detailed discussion in Sect. 4.5).

We added a brief summary of these results to Sect. 4.1, with references to previous works dealing with ROC segregation and preservation of viscous blobs in models with internal heating (lines 403-409, markup file).

**- I did not see the video supplement. Not sure if that's a problem on my end; I just want to make sure it's included in the final version that will be published.**

We ensured the video supplement (line 600, markup file) contains a working link to the online repository where the videos are stored.

In addition to the changes listed above, we:

1. propose a simpler title, for the sake of clarity: "Primordial-material preservation and Earth lower-mantle structure: the influence of recycled oceanic crust";
2. replaced: "vigor" --> "vigour"
3. fixed minor text inconsistencies, repetitions, typos etc. (lines 158, 170, 171, 176, 239, 359, 361, 412 in markup file; Caption Fig. 6 );
4. included additional acknowledgments (lines 690-692, see markup file)

We hope all these changes will improve the clarity of the manuscript, and thank again the Editor and Reviewers for their suggestions in this regard.

Best regards,

The Authors